# NleB2 from enteropathogenic *Escherichia coli* is a novel arginine-glucose transferase effector

Cristina Giogha[1,2], Nichollas E. Scott[3], Tania Wong Fok Lung[3], Georgina L. Pollock[1,2], Marina Harper[4], Ethan D. Goddard-Borger[5,6], Jaclyn S. Pearson[1,2,4], Elizabeth L. Hartland[1,2]*

1 Centre for Innate Immunity and Infectious Diseases, Hudson Institute of Medical Research, Clayton, Victoria, Australia, 2 Department of Molecular and Translational Science, Monash University, Clayton, Victoria, Australia, 3 Department of Microbiology and Immunology, University of Melbourne at the Peter Doherty Institute for Infection and Immunity, Melbourne, Victoria, Australia, 4 Department of Microbiology, Infection and Immunity Program, Monash Biomedicine Discovery Institute, Monash University, Clayton, Victoria, Australia, 5 ACRF Chemical Biology Division, The Walter and Eliza Hall Institute of Medical Research, Parkville, Victoria, Australia, 6 Department of Medical Biology, University of Melbourne, Parkville, Victoria, Australia

☯ These authors contributed equally to this work.
¤ Current address: Department of Pediatrics, Vagelos College of Physicians and Surgeons, Columbia University New York, New York, United States of America
* elizabeth.hartland@hudson.org.au

**Data Availability Statement:** All mass spectrometry proteomics data have been deposited to the ProteomeXchange Consortium via the PRIDE partner repository with the dataset identifier PXD021796 (In vitro RIPK1DD and NleB2

## Abstract

During infection, enteropathogenic *Escherichia coli* (EPEC) and enterohaemorrhagic *E. coli* (EHEC) directly manipulate various aspects of host cell function through the translocation of type III secretion system (T3SS) effector proteins directly into the host cell. Many T3SS effector proteins are enzymes that mediate post-translational modifications of host proteins, such as the glycosyltransferase NleB1, which transfers a single *N*-acetylglucosamine (GlcNAc) to arginine residues, creating an Arg-GlcNAc linkage. NleB1 glycosylates death-domain containing proteins including FADD, TRADD and RIPK1 to block host cell death. The NleB1 paralogue, NleB2, is found in many EPEC and EHEC strains but to date its enzymatic activity has not been described. Using *in vitro* glycosylation assays combined with mass spectrometry, we found that NleB2 can utilize multiple sugar donors including UDP-glucose, UDP-GlcNAc and UDP-galactose during glycosylation of the death domain protein, RIPK1. Sugar donor competition assays demonstrated that UDP-glucose was the preferred substrate of NleB2 and peptide sequencing identified the glycosylation site within RIPK1 as Arg603, indicating that NleB2 catalyses arginine glucosylation. We also confirmed that NleB2 catalysed arginine-hexose modification of Flag-RIPK1 during infection of HEK293T cells with EPEC E2348/69. Using site-directed mutagenesis and *in vitro* glycosylation assays, we identified that residue Ser252 in NleB2 contributes to the specificity of this distinct catalytic activity. Substitution of Ser252 in NleB2 to Gly, or substitution of the corresponding Gly255 in NleB1 to Ser switches sugar donor preference between UDP-GlcNAc and UDP-glucose. However, this switch did not affect the ability of the NleB variants to inhibit inflammatory or cell death signalling during HeLa cell transfection or EPEC infection. NleB2

glycosylation); PXD025057 (RIPK1 glycosylation infection assays) and PXD025531 (Confirmation of Arg-glucosylation of other death domain proteins).

**Funding:** This work was funded by the National Health and Medical Research Council of Australia (NHMRC) applications APP1098826 and APP1175976 awarded to ELH and applications APP1100164 and APP1037373 awarded to NES. Salary of NES was supported by APP1037373 and FT200100270. Salary of CG was supported by APP1175976. Salary of JSP was supported by APP1159230. The work was additionally supported by a Victoria Fellowship awarded to CG. The funders had no role in study design, data collection and analysis, decision to publish, or preparation of the manuscript.

**Competing interests:** The authors have declared that no competing interests exist.

is thus the first identified bacterial Arg-glucose transferase that, similar to the NleB1 Arg-GlcNAc transferase, inhibits host protein function by arginine glycosylation.

## Author summary

Bacterial gut pathogens including enteropathogenic *E. coli* (EPEC) and enterohaemorrhagic *E. coli* (EHEC), manipulate host cell function by using a type III secretion system to inject 'effector' proteins directly into the host cell cytoplasm. We and others have shown that many of these effectors are novel enzymes, including NleB1, which transfers a single *N*-acetylglucosamine (GlcNAc) sugar to arginine residues, mediating Arg-GlcNAc glycosylation. Here, we found that a close homologue of NleB1 that is also present in EPEC and EHEC termed NleB2, uses a different sugar during glycosylation. We demonstrated that in contrast to NleB1, the preferred nucleotide-sugar substrate of NleB2 is UDP-glucose and we identified the amino acid residue within NleB2 that dictates this unique catalytic activity. Substitution of this residue in NleB2 and NleB1 switches the sugar donor usage of these enzymes but does not affect their ability to inhibit host cell signalling. Thus, NleB2 is the first identified bacterial arginine-glucose transferase, an activity which has previously only been described in plants and algae.

## Introduction

Diarrhoeagenic *Escherichia coli* including enteropathogenic *E. coli* (EPEC) and enterohaemorrhagic *E. coli* (EHEC), cause nearly 200,000 deaths annually worldwide [1]. EPEC and EHEC remain extracellular during infection and use a type III secretion system (T3SS) to translocate 'effector' proteins into the host cell cytoplasm [2]. These effectors orchestrate control over host cell physiology to facilitate infection and allow the bacteria to evade innate immune defence mechanisms [3]. Recent work has revealed that several EPEC and EHEC effectors are novel enzymes that target and interfere with the function of host cell signalling proteins. For example, we and others discovered that NleB1 is a novel glycosyltransferase that modifies human death domain-containing proteins including FADD, TRADD and RIPK1, which are essential components of death-receptor signalling [4,5]. The enzymatic activity of NleB1 is unusual, as it utilizes the sugar donor UDP-*N*-acetylglucosamine (UDP-GlcNAc) to glycosylate arginine residues in target proteins, forming an arginine-GlcNAc (Arg-GlcNAc) linkage. The addition of a single GlcNAc moiety to Arg residues in death domain proteins blocks the formation of critical immune signalling complexes, resulting in inhibition of inflammation and cell death during infection.

Homologues of NleB1 are found in *Citrobacter rodentium* (termed NleB) and *Salmonella* species (termed SseK1, SseK2, SseK3) [6–8]. These homologues have well-characterized roles in inhibition of death receptor signalling [4,5,9–15] and some also block small Rab GTPase function [16,17]. However, EHEC and EPEC strains also contain the NleB1 paralogue NleB2, which is less well-characterized. NleB2 inhibits NF-κB signalling in transfected cells similarly to NleB1 [4], but does not appear to inhibit cell death signalling during EPEC infection [5]. Furthermore, previous studies reported that compared to NleB1, NleB2 had a lower Arg-GlcNAc enzymatic activity towards TRADD in transfected cells, and no Arg-GlcNAcylation of TRADD was detected by immunoblot upon incubation with NleB2 *in vitro* [4,18].

Here we characterized the glycosylation mediated by NleB2. We found that NleB2 can utilize either UDP-GlcNAc, UDP-glucose or UDP-galactose to glycosylate Arg603 within the

death domain of RIPK1. However, competition assays defined the preferred sugar donor for NleB2 as UDP-glucose, and this specificity was dependent on Ser252 in NleB2, which corresponds to Gly255 in NleB1. Substitution of this residue switched sugar donor preference from UDP-glucose to UDP-GlcNAc in NleB2, and vice versa in NleB1, but this did not affect inhibition of death receptor signalling in transfected cells or during EPEC infection.

## Results

### Interaction of NleB2 with the death domains of RIPK1 and TNFR1

Given that NleB1 targets the death domains of FADD, TRADD, RIPK1 and TNFR1 and shares 84% similarity with NleB2 at the amino acid level, we tested whether NleB2 also interacted with death domain containing proteins. Using a yeast-two-hybrid pairwise interaction system [19], we found that NleB2 interacted with the death domains of human RIPK1 and TNFR1 similar to NleB1 (Fig 1A), but we could not detect interactions with the death domains of TRADD, FADD and FAS (not shown). To assess whether NleB2 inhibited TNFR1 and RIPK1-mediated signalling we performed NF-κB-dependent luciferase assays on HeLa cells stimulated with TNF. As observed previously [4,20], NleB1 inhibited TNF-induced NF-κB activation under these conditions, and this inhibition was dependent on NleB1 glycosyltransferase activity, as the catalytic mutant GFP-NleB1$_{DXD}$ did not inhibit NF-κB activation (Fig 1B). NleB2 also inhibited TNF-induced NF-κB activation, although not as potently as NleB1 (Fig 1B). The catalytic activity of NleB2 was also required for inhibition of the NF-κB pathway (Fig 1B).

### NleB2 from EPEC does not GlcNAcylate arginine in a cellular environment

Our observation that the NleB2 DXD motif was required for inhibition of NF-κB signaling suggested that NleB2 may glycosylate and inactivate TNFR1 or RIPK1 to block NF-κB

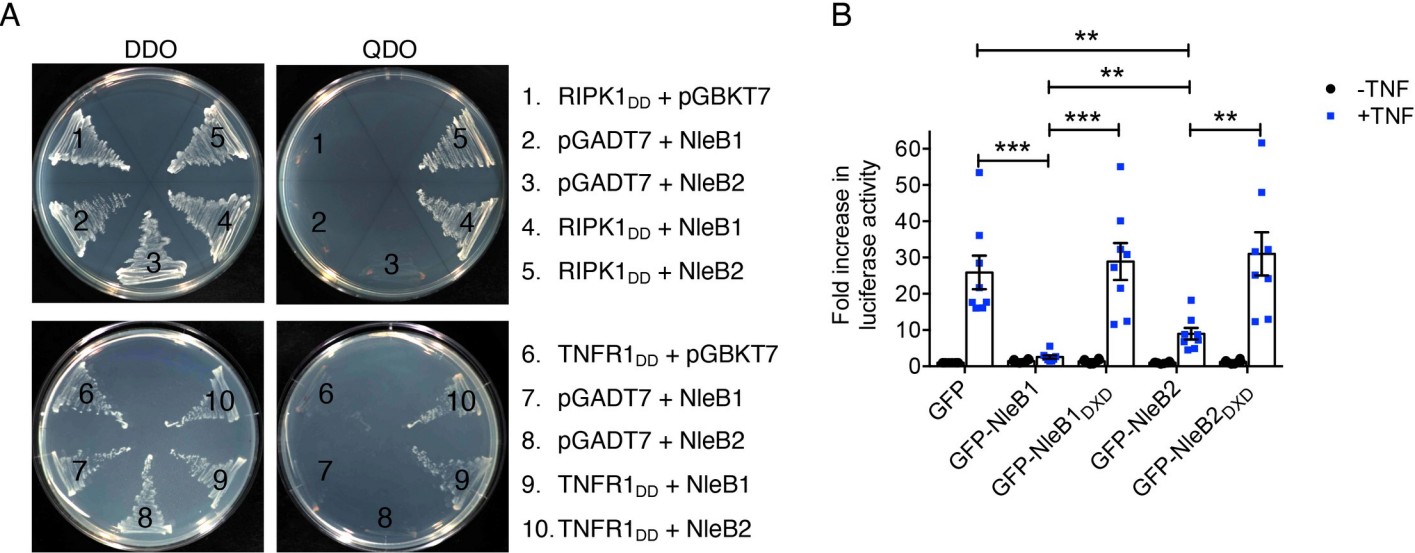

**Fig 1. Binding of NleB2 to the death domains of RIPK1 and TNFR1 and inhibition of NF-κB signalling. (A)** Yeast-2-hybrid analysis of protein-protein interactions in *S. cerevisiae* PJ69-4A. Growth on selective media for plasmid maintenance (DDO, double dropout) or selective media for interaction between proteins (QDO, quadruple dropout). **(B)** Fold increase in NF-κB-dependent luciferase activity in HeLa cells transfected with pEGFP-C2, pGFP-NleB1, pGFP-NleB1$_{DXD}$, pGFP-NleB2 or pGFP-NleB2$_{DXD}$ and either left unstimulated or stimulated with TNF for 16 h where indicated. Results are the mean ± standard error of the mean (SEM) of three independent experiments carried out in duplicate. ${}^{**}$p < 0.01, ${}^{***}$p < 0.001, unpaired, two-tailed *t*-test.

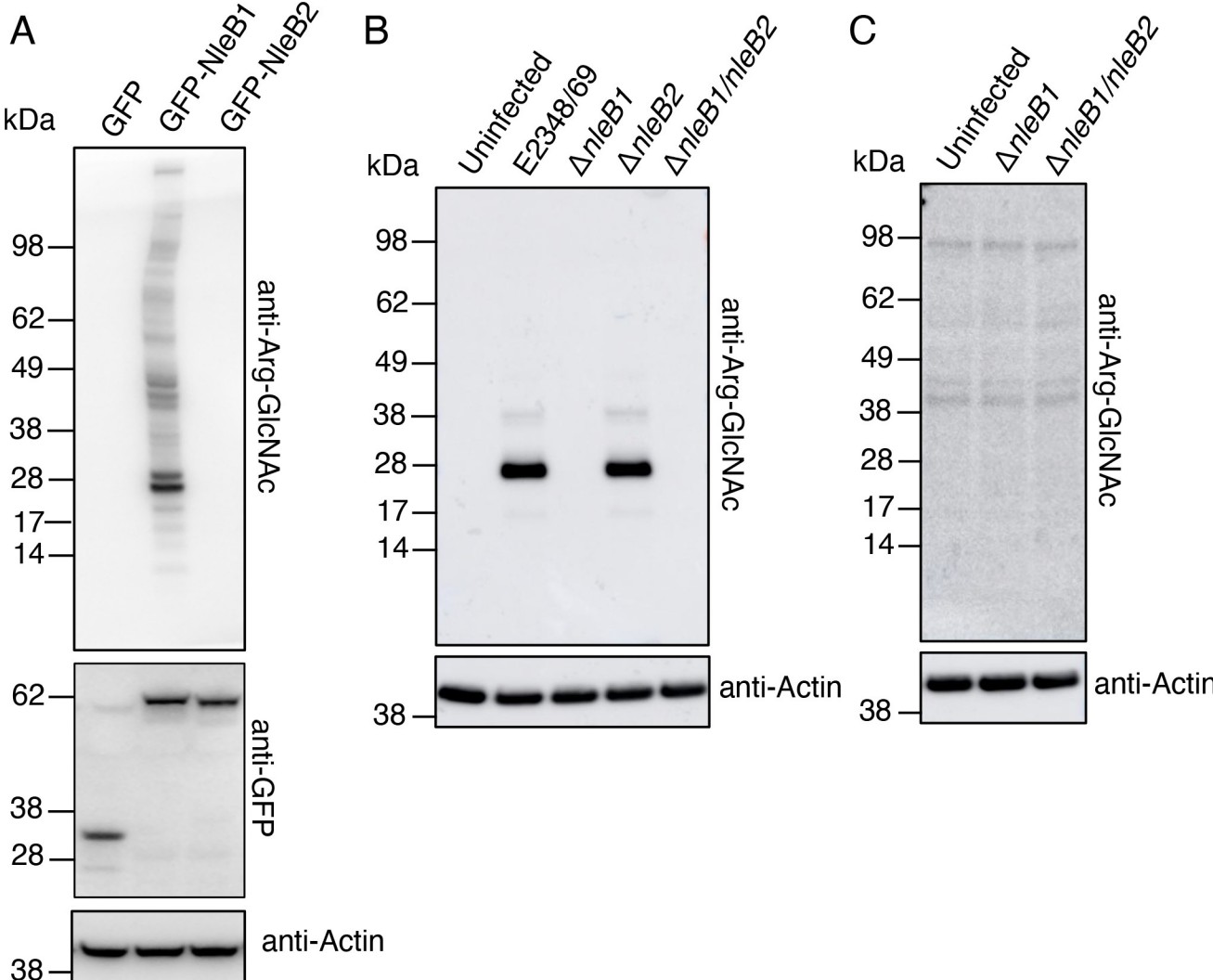

**Fig 2. Lack of Arg-GlcNAcylation by NleB2 upon ectopic expression or translocation into host cells during infection. (A)** HEK293T cells were transfected with either pEGFP-C2, pGFP-NleB1 or pGFP-NleB2 before being lysed and subjected to polyacrylamide gel electrophoresis (PAGE) and immunoblot with anti-Arg-GlcNAc and anti-GFP antibodies. Anti-Actin antibodies were used as a loading control. Representative immunoblots of at least 3 independent experiments. **(B)** HT-29 cells were either left uninfected or infected with EPEC E2348/69 or derivatives lacking *nleB1* and/or *nleB2* for 3 hours before being lysed and subjected to PAGE and immunoblot with anti-Arg-GlcNAc antibodies. Anti-Actin antibodies were used as a loading control. Representative immunoblots of at least 3 independent experiments. **(C)** HT-29 cells were either left uninfected or infected with EPEC Δ*nleB1* or EPEC Δ*nleB1/nleB2* for 3 hours before being lysed and subjected to PAGE and immunoblot with anti-Arg-GlcNAc antibodies. Anti-Actin antibodies were used as a loading control. Representative immunoblots of at least 3 independent experiments.

activation. To examine NleB2 glycosylation in human cells we expressed either GFP, GFP-NleB1 or GFP-NleB2 ectopically in HEK293T cells and subjected the cell lysates to immunoblot using anti-Arg-GlcNAc antibodies [21]. While expression of GFP-NleB1 induced Arg-GlcNAcylation of multiple proteins, no significant Arg-GlcNAcylation was detected in HEK293T cells expressing NleB2 (Fig 2A). We then examined the activity of NleB2 during EPEC infection of HT-29 cells. Using anti-Arg-GlcNAc antibodies we detected a prominent band around 28 kDa in size and two additional bands of 38 kDa and 17 kDa in size during infection with wild type EPEC E2348/69 (Fig 2B). These Arg-GlcNAc modified proteins were not detected during infection with strains lacking *nleB1* including EPEC Δ*nleB1* and EPEC

$\Delta nleB1/nleB2$ and were unaffected by deletion of *nleB2* alone (Fig 2B). Therefore, the Arg-GlcNAcylated proteins detected by immunoblot during wild type EPEC infection of HT-29 cells were due to the presence of NleB1. Furthermore, no Arg-GlcNAcylated proteins were detected during EPEC $\Delta nleB1$ infection of HT-29 cells even upon longer exposure (Fig 2C). These results support the conclusion that NleB2 does not mediate Arg-GlcNAcylation.

## Nucleotide-sugar utilization by NleB2

It is well known that glycosyltransferase nucleotide-sugar donor substrate specificity can be completely altered by point mutations [22–24]. We hypothesized that the lack of NleB2-me-diated Arg-GlcNAcylation detected may be due to the preference of NleB2 for a nucleotide-sugar other than UDP-GlcNAc. Common sugar donors found in the cytoplasm of mammalian cells include UDP-glucose, UDP-galactose, UDP-*N*-acetylgalactosamine (UDP-GalNAc), UDP-glucuronic acid (UDP-GlcA) and GDP-mannose [25]. To explore whether these sugar donors were utilised by NleB2, we performed *in vitro* glycosylation assays using the purified death domain of RIPK1. Intact protein analysis by liquid chromatography-mass spectrometry (LC-MS) revealed that the mass of MBP-RIPK1$_{DD}$ was shifted by an amount commensurate with a single sugar modification in the presence of GST-NleB2 and either 50 µM UDP-GlcNAc (203 Da), UDP-glucose (162 Da) or UDP-galactose (162 Da) but not in the presence of UDP-GlcA, UDP-GalNAc or GDP-mannose (Fig 3). We detected mass shifts in both the full length (~53068 Da) and truncated proteoform (~51707 Da) of MBP-RIPK1$_{DD}$ (Fig 3), which did not occur in the presence of catalytically inactive GST-NleB2$_{DXD}$ (S1 Fig).

## UDP-glucose utilization and arginine glycosylation by NleB2

We next sought to understand if NleB2 preferentially utilized any of the three transferable nucleotide-sugars *in vitro*. We titrated the concentration of sugar donors to 0.5 µM and short-ened the incubation period to 20 minutes rather than 3 hours. Under these conditions, only UDP-glucose was utilised by GST-NleB2 to glycosylate MBP-RIPK1$_{DD}$ (S2 Fig). To further assess NleB2 sugar donor preference, we performed an *in vitro* nucleotide-sugar competition assay to compare UDP-GlcNAc with either UDP-glucose or UDP-galactose. Using intact pro-tein LC-MS we detected only GlcNAc-modified MBP-RIPK1$_{DD}$ when incubated in the pres-ence of NleB2 and both UDP-GlcNAc and UDP-galactose, suggesting UDP-GlcNAc is the preferred sugar donor under these conditions (Fig 4A). In contrast, we detected mass shifts corresponding to both GlcNAc and glucose modification of MBP-RIPK1$_{DD}$ when UDP-Glc-NAc and UDP-glucose were co-incubated in the presence of NleB2 (Fig 4A). Glucose-modi-fied MBP-RIPK1$_{DD}$ was approximately 4 times more abundant than GlcNAc-modified MBP-RIPK1$_{DD}$, supporting the preference of NleB2 for UDP-glucose as the donor (Fig 4A). Although glucose and galactose modifications are indistinguishable by mass and were there-fore not directly compared, it is likely that UDP-glucose is the preferred sugar donor overall among UDP-glucose, UDP-galactose and UDP-GlcNAc, as modification of MBP-RIPK1$_{DD}$ by NleB2 in the presence of using UDP-galactose was comparatively low (Fig 3), and modification of MBP-RIPK1$_{DD}$ with galactose did not occur when UDP-GlcNAc was present (Fig 4A). We observed similar results by immunoblot using anti-Arg-GlcNAc antibodies, which confirmed that UDP-glucose but not UDP-galactose successfully competed with UDP-GlcNAc resulting in reduced Arg-GlcNAcylation of MBP-RIPK1$_{DD}$ by NleB2 (Fig 4B).

To identify the site of Arg-glucose modification, *in vitro* glucosylated MBP-RIPK1$_{DD}$ was digested with Glu-C and subjected to LC-MS/MS with EThcD fragmentation, revealing the glucosylated residue was Arg603 within the death domain of RIPK1 (S3A Fig). Interestingly, we also observed dual glucosylation of Arg603 by NleB2 (S3A Fig), likely due to the high

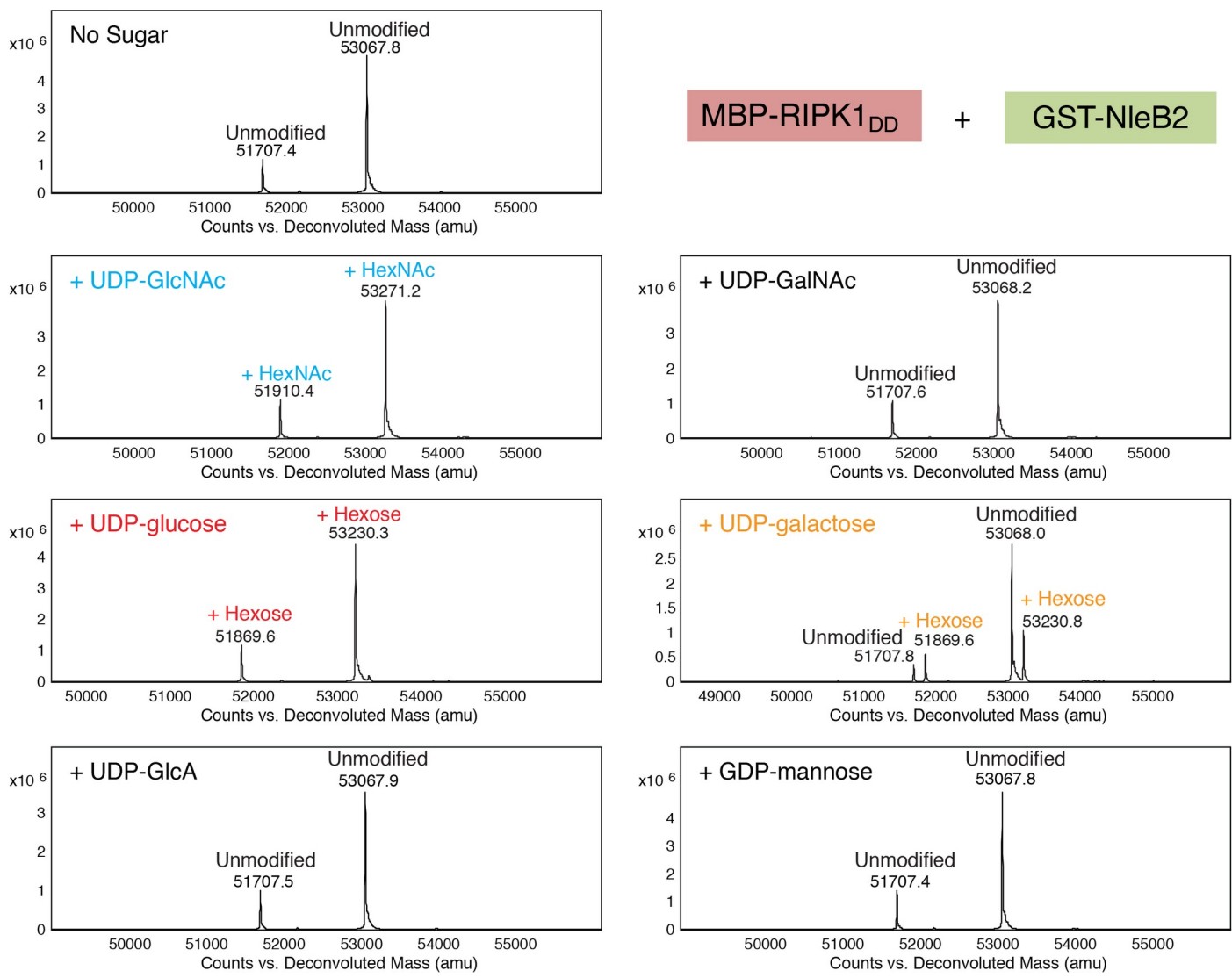

**Fig 3. Utilization of UDP-GlcNAc, UDP-glucose and UDP-galactose by NleB2.** Deconvoluted intact mass spectra of MBP-RIPK1$_{DD}$ proteoforms (full-length expected average mass 53081 Da) incubated with GST-NleB2 either without sugar donors, or in the presence of one of UDP-GlcNAc, UDP-glucose, UDP-GalNAc, UDP-galactose, UDP-glucuronic acid or GDP-mannose at 50 μM.

concentration of UDP sugars (10 mM) that we used in this assay to promote complete modification of RIPK1$_{DD}$. Cytoplasmic UDP-sugar concentrations in eukaryotic cells are estimated to be in the micromolar range [26]. Intact mass spectrometry and peptide analysis further supported dual modification of RIPK1 by NleB2 in the presence of 10 mM UDP-glucose but not UDP-GlcNAc or UDP-galactose (S3B and S4 Figs). The modification site was confirmed by mutation of Arg603 to alanine, which abolished NleB2-mediated GlcNAc, glucose or galactose modifications *in vitro*, as observed by intact mass spectrometry (S5A Fig), anti-Arg-GlcNAc immunoblot (S5B Fig) and peptide analysis (S5C–S5E Fig).

Arg603 in RIPK1 corresponds to an arginine that is conserved in many death domains and is also targeted for GlcNAcylation by NleB1 [4]. Therefore, we examined whether NleB2 could catalyse arginine-glucose modification of other death domain-containing proteins. Peptide analysis of *in vitro* glycosylation assays revealed that NleB2 modified the conserved arginine within purified FADD (Arg117) but did not modify the death domains of FAS or TRAILR2

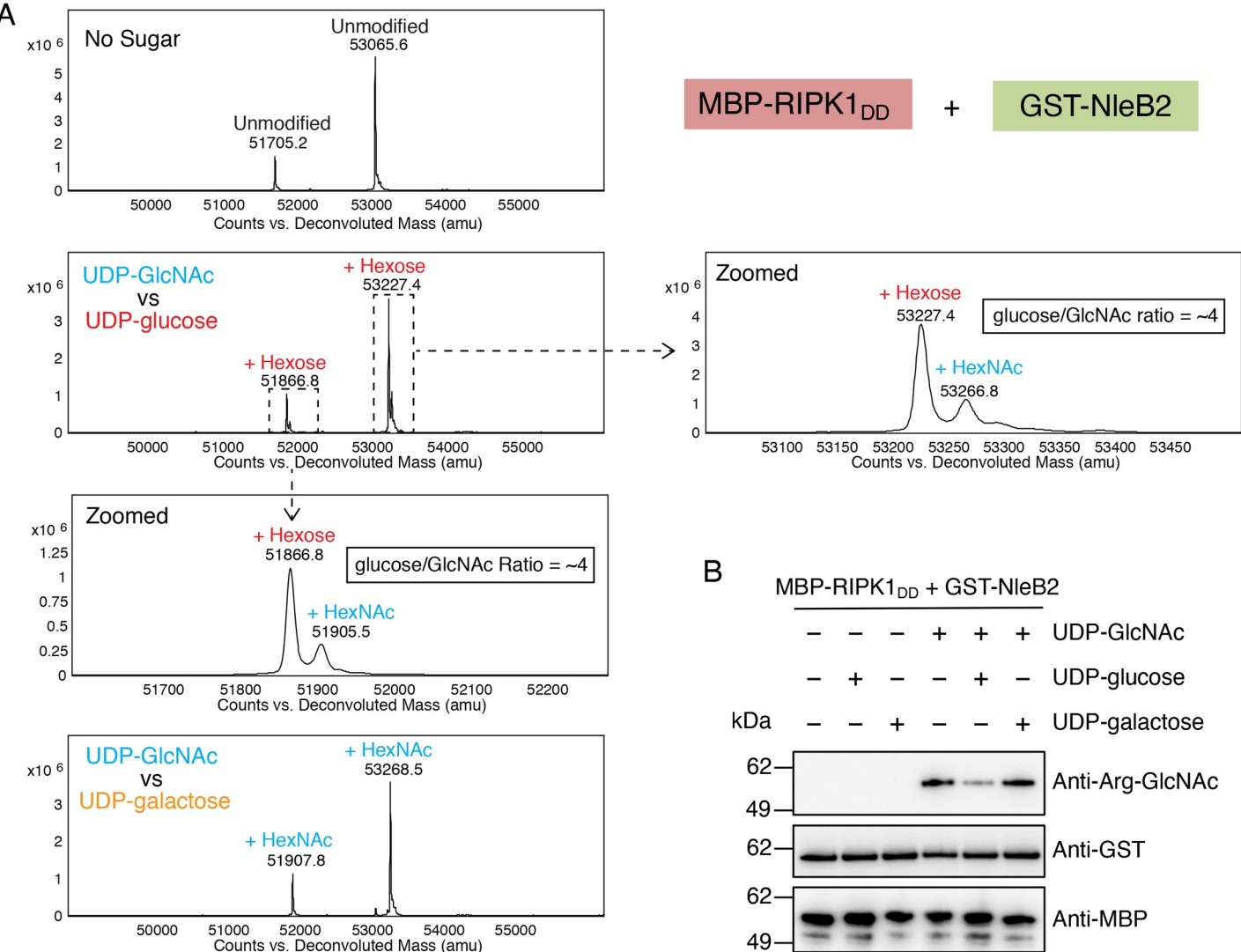

**Fig 4. Preference of NleB2 for UDP-glucose *in vitro*. (A)** Deconvoluted intact mass spectra of *in vitro* sugar donor competition assays. MBP-RIPK1$_{DD}$ was incubated with GST-NleB2 without sugar donors, or in the presence of UDP-GlcNAc and either UDP-galactose or UDP-glucose at 25 μM concentrations of each sugar donor. **(B)** *In vitro* sugar donor competition assays. MBP-RIPK1$_{DD}$ was incubated with GST-NleB2 without sugar donors, or in the presence of 25 μM UDP-glucose, UDP-galactose or UDP-GlcNAc individually or in combination. Proteins were subjected to PAGE and immunoblot with anti-ArgGlcNAc, or anti-MBP and anti-GST antibodies as controls. Representative of at least 3 experiments.

when incubated in the presence of UDP-glucose (S6A Fig and S1 Table). As the death domains of TRADD and TNFR1 are insoluble when expressed in *E. coli*, we examined whether NleB2 could glucosylate Flag-TRADD or Flag-TNRF1$_{DD}$ in co-transfected HEK293T cells. We detected both hexose-modified and HexNAc-modified TNFR1 when expressed with catalytically active NleB2, and modification occurred on the conserved Arg376 (S6B Fig). We did not detect Arg-hexose or Arg-HexNAc-modified peptides when Flag-TRADD was co-expressed with NleB2 (S1 Table).

## Residue Ser252 within NleB2 decides sugar donor preference for UDP-glucose

To understand the basis of nucleotide-sugar preference for NleB1 and NleB2, we aligned the amino acid sequences of all known Arg-GlcNAc transferases including NleB1 and NleB2 from

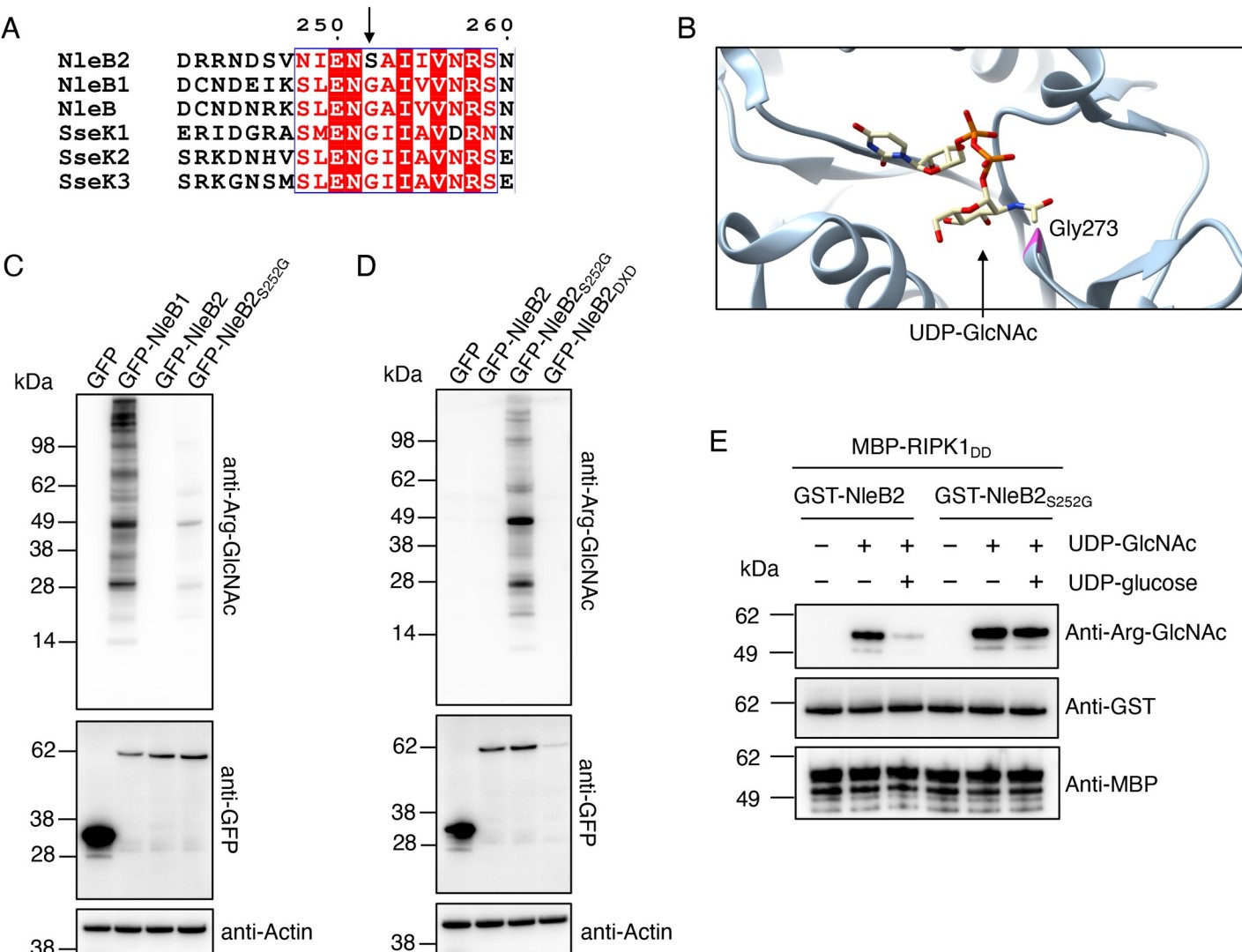

**Fig 5. Role of Serine 252 within NleB2 in preference for UDP-glucose. (A)** Alignment of NleB2 and NleB1 from EPEC O127:H6 strain E2348/69, NleB from *Citrobacter rodentium* strain ICC168 and SseK1, SseK2 and SseK3 from *Salmonella enterica* serovar Typhimurium strain SL1344. Arrow indicates serine 252 within NleB2. Alignment was performed using ClustalW and visualised using ESPript. **(B)** The crystal structure of SseK2 (PDB ID: 5H63)[18] in complex with UDP-GlcNAc [PMID: 30327479]. The peptide backbone is depicted as a blue ribbon, Glycine 273 is highlighted in pink, and the ligand represented in stick form. The image was generated using Chimera 1.14 with carbon in yellow, nitrogen in blue, oxygen in red and phosphorus in orange. **(C)** HEK293T cells were transfected with either pEGFP-C2, pGFP-NleB1, pGFP-NleB2 or pGFP-NleB2$_{S252G}$ before being lysed and subjected to PAGE and immunoblot with anti-Arg-GlcNAc and anti-GFP antibodies. Anti-Actin antibodies were used as a loading control. Representative immunoblots of at least 3 independent experiments. **(D)** HEK293T cells were transfected with either pEGFP-C2, pGFP-NleB2, pGFP-NleB2$_{S252G}$ or pGFP-NleB2$_{DXD}$ before being lysed and subjected to PAGE and immunoblot with anti-Arg-GlcNAc and anti-GFP antibodies. Anti-Actin antibodies were used as a loading control. Representative immunoblots of at least 3 independent experiments. **(E)** *In vitro* sugar donor competition assays. MBP-RIPK1$_{DD}$ was incubated with GST-NleB2 or GST-NleB2$_{S252G}$ either without sugar donors, or in the presence of 25 μM UDP-GlcNAc alone or in combination with 25 μM UDP-glucose. Proteins were subjected to PAGE and immunoblot with anti-ArgGlcNAc, or anti-MBP and anti-GST antibodies as controls. Representative of at least 3 experiments.

EPEC strain E2348/69, SseK1, SseK2 and SseK3 from *Salmonella enterica* serovar Typhimurium strain SL1344 and NleB from *Citrobacter rodentium* strain ICC168. Among other differences, we identified Ser252 in NleB2, which was a glycine residue in all other homologous sequences (Fig 5A). Of the known structures of NleB and SseK effectors, only SseK2 has been co-crystallised with intact UDP-GlcNAc [18]. Examination of the published structure in

complex with UDP-GlcNAc showed that Gly273 in SseK2, corresponding to Ser252 in NleB2, was located in the UDP-GlcNAc binding region, near the acetyl group of GlcNAc (Fig 5B).

To test if the amino acid residue at this position contributed nucleotide-sugar preference, we mutated Ser252 in NleB2 to glycine. Upon expression of GFP-NleB2$_{S252G}$ in HEK293T cells, we observed enhanced Arg-GlcNAcylation compared to GFP-NleB2 using anti-Arg-GlcNAc antibodies, although Arg-GlcNAcylation was notably more abundant in NleB1-transfected cells when immunoblots were imaged together (Fig 5C and 5D). This amino acid substitution did not affect the ability to utilize nucleotide-sugar donors when added individually, as GST-NleB2$_{S252G}$ could utilise either UDP-GlcNAc or UDP-glucose to modify RIPK1$_{DD}$ (S7A Fig). However, GST-NleB2$_{S252G}$ demonstrated a preference for UDP-GlcNAc when co-incubated with UDP-glucose in the presence of MBP-RIPK1$_{DD}$ (S8 Fig). Furthermore, co-incubation of UDP-glucose with UDP-GlcNAc had no effect on Arg-GlcNAcylation of MBP-RIPK1$_{DD}$ by NleB2$_{S252G}$ as detected by immunoblot (Fig 5E), suggesting this mutant did not utilize UDP-glucose to any significant extent when UDP-GlcNAc was present.

Overall, our data supported that mutation of Ser252 to glycine in NleB2 changed the sugar donor preference to that of NleB1, which was previously shown to utilize UDP-GlcNAc over any of UDP-glucose, UDP-GalNAc, UDP-galactose or UDP-GlcA [4]. Here we confirmed NleB1 preferred UDP-GlcNAc over UDP-glucose (S8 Fig). However, we also observed that NleB1 utilized UDP-glucose *in vitro* when no other sugar donors were present (S7B Fig). Hence, there is a certain amount of fluidity in nucleotide-sugar utilization by the NleB/SseK family of arginine glycosyl transferases.

## Mutation of Gly255 to serine within NleB1 switches sugar donor preference to UDP-glucose but does not affect cellular function

To determine whether the residue corresponding to Ser252 in NleB2 played a role in the substrate preference of NleB1 for UDP-GlcNAc, we substituted Gly255 in NleB1 with serine. We found that expression of GFP-NleB1$_{G255S}$ in HEK293T cells resulted in significantly reduced Arg-GlcNAcylation of host proteins when compared to cells expressing GFP-NleB1 (Fig 6A). A weakly Arg-GlcNAcylated protein of approximately 28 kDa in size was detected in GFP-NleB1$_{G255S}$-expressing HEK293T cells only when GFP-NleB1 or GFP-NleB2$_{S252G}$ expressing cells were not included on the same immunoblot, and the immunoblot was developed with a more sensitive chemiluminescent substrate (Fig 6B). This band may represent Arg-GlcNAc modified FADD, given that NleB1 preferentially targets FADD when expressed at native levels [27]. *In vitro* glycosylation assays showed that NleB1$_{G255S}$ was functional, and utilised either UDP-GlcNAc or UDP-glucose to modify MBP-RIPK1$_{DD}$ (S7C Fig). In contrast to NleB1, UDP-glucose was preferred by NleB1$_{G255S}$ to modify MBP-RIPK1$_{DD}$ when both UDP-GlcNAc and UDP-glucose were co-incubated (S8 Fig). Similar results were observed using His-FADD, which is the major target of NleB1 during EPEC infection [27] (Fig 6C). Co-incubation with UDP-glucose caused a reduction in Arg-GlcNAc modification of His-FADD by NleB1$_{G255S}$ as detected by immunoblot but did not affect NleB1-mediated Arg-GlcNAcylation of His-FADD (Fig 6C), indicating that Gly255 was central to the preference of NleB1 for UDP-GlcNAc over UDP-glucose when both sugar donors were present.

To assess potential differences in the enzyme kinetics of NleB1, NleB2 and site-directed mutants we performed UDP-Glo assays and measured the rate of UDP release during *in vitro* glycosylation reactions using conditions similar to those previously optimised for NleB1 [28]. We performed the assay using 150 nM GST-NleB1 which was incubated in the presence of UDP-GlcNAc alone. We observed low levels of UDP release with increasing concentrations of UDP-GlcNAc (S9A Fig). However, the release of UDP was considerably more efficient upon

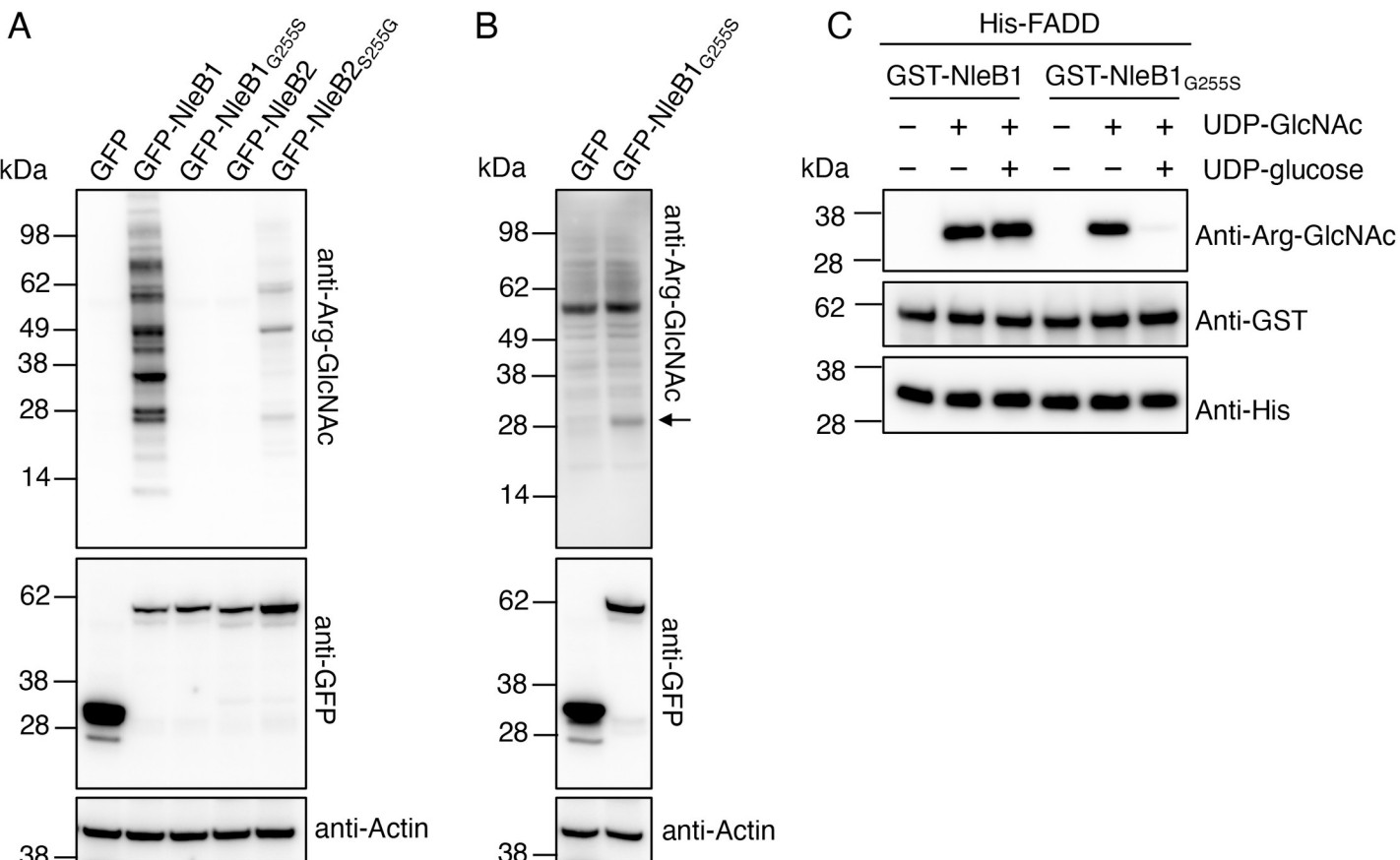

**Fig 6. Mutation of Glycine 255 in NleB1 switches sugar donor preference to UDP-glucose. (A)** HEK293T cells were transfected with either pEGFP-C2, pGFP-NleB1, pGFP-NleB1$_{G255S}$, pGFP-NleB2 or pGFP-NleB2$_{S252G}$ before being lysed and subjected to PAGE and immunoblot with anti-Arg-GlcNAc and anti-GFP antibodies. Anti-Actin antibodies were used as a loading control. Anti-Arg-GlcNAc immunoblot was developed with routine detection reagents. Representative immunoblots of 3 independent experiments. **(B)** HEK293T cells were transfected with either pEGFP-C2 or pGFP-NleB1$_{G255S}$ before being lysed and subjected to PAGE and immunoblot with anti-Arg-GlcNAc and anti-GFP antibodies. Anti-Actin antibodies were used as a loading control. Anti-Arg-GlcNAc immunoblot was developed with highly sensitive detection reagents. Representative immunoblots of 3 independent experiments. **(C)** *In vitro* sugar donor competition assays. His-FADD was incubated with GST-NleB1 or GST-NleB1$_{G255S}$ either without sugar donors, or in the presence of 25 μM UDP-GlcNAc alone or in combination with 25 μM UDP-glucose. Proteins were subjected to PAGE and immunoblot with anti-Arg-GlcNAc, or anti-His and anti-GST antibodies as controls. Representative of at least 3 experiments.

co-incubation with 1 μM MBP-RIPK1$_{DD}$ (S9A Fig). Therefore, we performed all enzymatic kinetic analysis and Km calculations on reactions that included MBP-RIPK1$_{DD}$ as a protein target (S9B and S9C Fig). We found that NleB1 and NleB1$_{G255S}$ were generally more efficient at utilising either UDP-linked sugar compared to NleB2 and NleB2$_{S252G}$ under the conditions tested, with higher Vmax and lower Km values determined overall (S9 and S9C Fig). As expected, we observed no UDP release in the reactions using catalytically inactive NleB1$_{DXD}$ or NleB2$_{DXD}$ (S9B Fig). Km and Vmax calculations performed using Michaelis-Menten equations confirmed that NleB2 and NleB1$_{G255S}$ used UDP-glucose more efficiently than UDP-GlcNAc, while the reverse was true for NleB2$_{S252G}$ and NleB1 (S9C Fig). Interestingly, the enzyme that utilised UDP-glucose most efficiently under the conditions tested was NleB1$_{G255S}$, which had a Km of 4.5 μM with UDP-glucose, compared to NleB2 which was determined to have a Km of 32.9 μM with UDP-glucose (S9C Fig). It is possible that other residues in NleB2 influence the target binding affinity of NleB2 for MBP-RIPK1$_{DD}$ compared to NleB1. Such differences in protein target specificity may explain the differing enzymatic efficiencies between NleB1 and NleB2 in the conditions tested here.

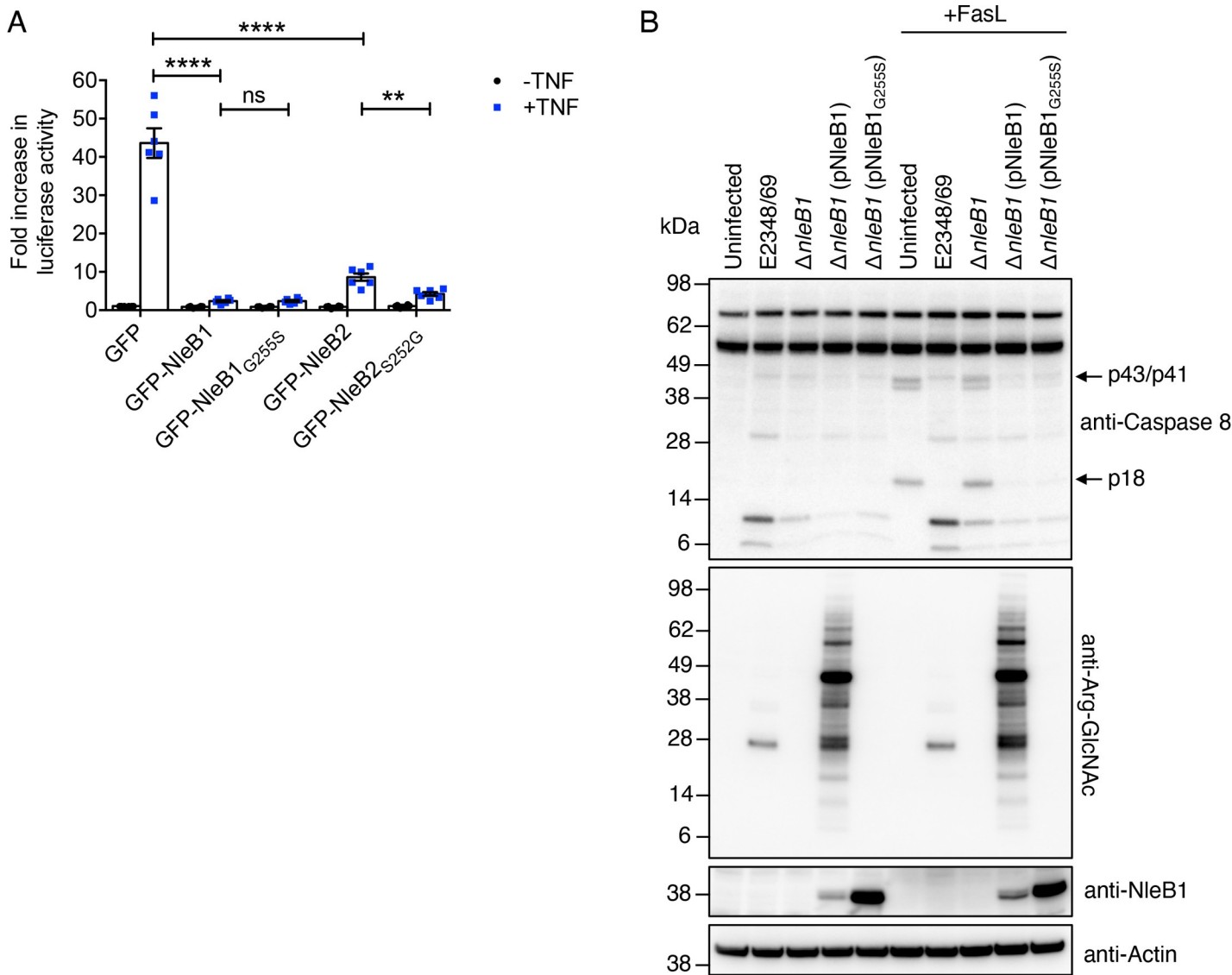

**Fig 7. Impact of Arg-glucose post-translational modification on protein function. (A)** Fold increase in NF-κB-dependent luciferase activity in HeLa cells transfected with pEGFP-C2, pGFP-NleB1, pGFP-NleB1$_{G255S}$, pGFP-NleB2 or pGFP-NleB2$_{S252G}$ and either left unstimulated or stimulated with TNF for 16 h where indicated. Results are the mean ± standard error of the mean (SEM) of three independent experiments carried out in duplicate. $^{**}p < 0.01$, $^{****}p < 0.0001$, ns (not significant), unpaired, two-tailed $t$-test. **(B)** HT-29 cells were either left uninfected or infected with EPEC E2348/69 or derivatives for 3 hours before being stimulated with 20 ng/ml FasL for 2 hours and then lysed and subjected to PAGE and immunoblot with anti-caspase-8 and anti-Arg-GlcNAc antibodies. Anti-Actin antibodies were used as a loading control. Representative immunoblots of at least 3 independent experiments.

To understand whether changing the UDP-sugar preference of NleB1 or NleB2 affected the cellular function of the enzymes, we performed NF-κB-dependent luciferase assays. We found that both NleB1$_{G255S}$ and NleB2$_{S252G}$ inhibited NF-κB activation in response to TNF stimulation in transfected HeLa cells similar to the parent enzymes (Fig 7A). During infection, NleB1$_{G255S}$ inhibited caspase-8 cleavage in response to FasL stimulation and complemented EPEC Δ*nleB1* similarly to NleB1 (Fig 7B). However, unlike NleB1, NleB1$_{G255S}$ did not Arg-GlcNAcylate host proteins during infection (Fig 7B). Thus, the Arg-glucose modification of FADD by NleB1$_{G255S}$ appeared to inhibit death receptor signaling similar to Arg-GlcNAc modification by NleB1.

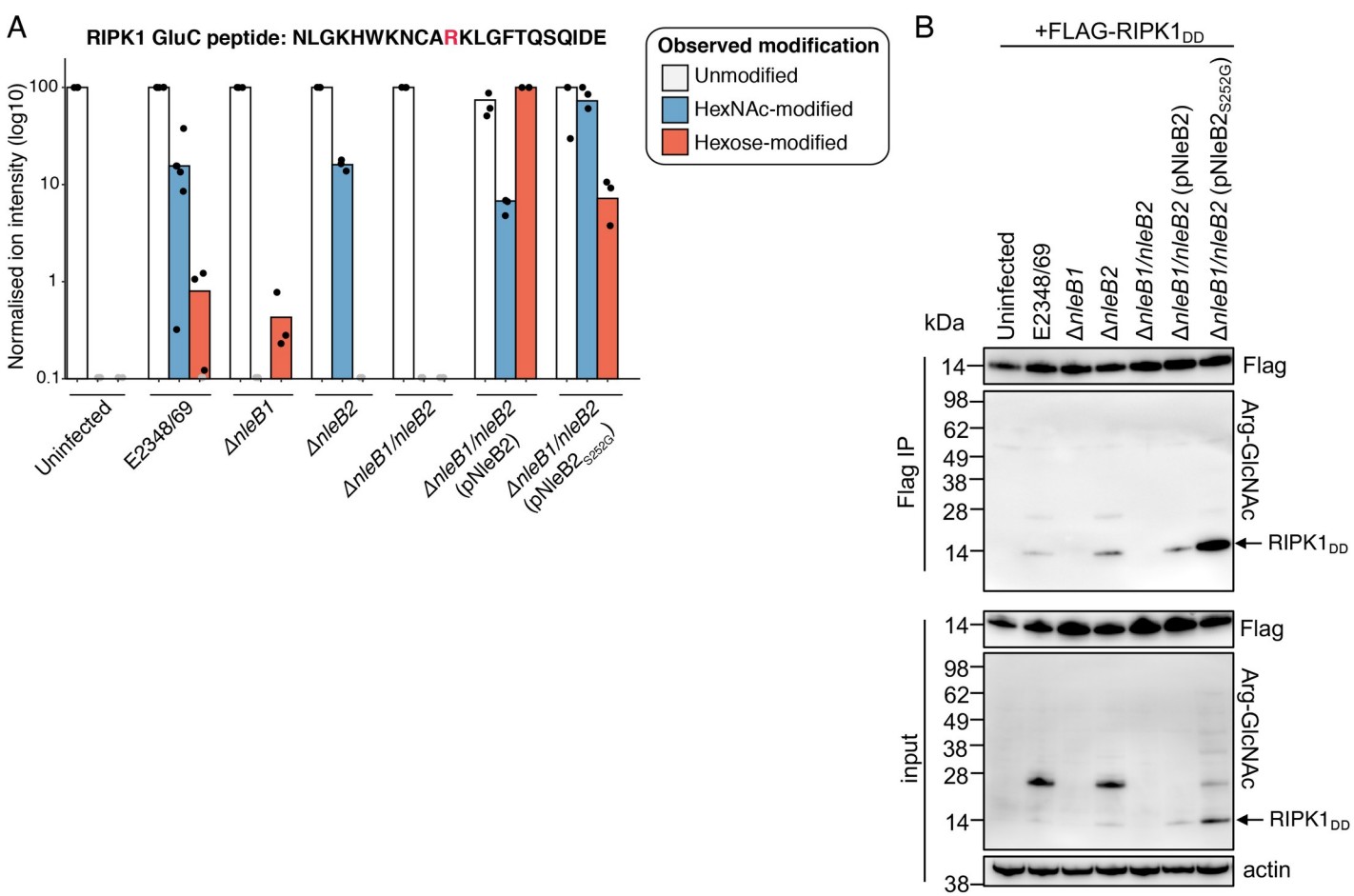

**Fig 8. Arginine glucosylation of RIPK1 during EPEC infection.** HEK293T cells were transfected with pFlag-RIPK1$_{DD}$ before being infected with derivatives of EPEC E2348/69 for 3 hours. Anti-Flag immunoprecipitation was performed on cell lysates and eluates were subjected to mass spectrometry **(A)** or immunoblot analysis **(B)**. **(A)** Normalised ion intensities of the peptide forms of NLGKHWKNCARKLGFTQSQIDE identified in Flag-RIPK1$_{DD}$ immunoprecipitated from EPEC-infected HEK293T cells. Results are shown as log10 of ion intensities normalised to the most abundant peptide form identified in each condition. Black points represent the peptides detected in each replicate, and grey points indicate that the peptide was not detected within the replicate. Combined data from at least 3 independent experiments for each condition. **(B)** Immunoblot analysis of input and eluate (Flag IP) of Flag-RIPK1$_{DD}$ immunoprecipitated from EPEC-infected HEK293T cells. Membranes were probed with anti-Flag or anti-Arg-GlcNAc. Anti-Actin antibodies were used as a loading control. Representative of at least 3 independent experiments.

## Arginine glucosylation of RIPK1 during EPEC infection

Given that both NleB1 and NleB2 glycosylate Arg603 within the death domain of RIPK1 (S3A Fig) [4], we investigated arginine modification of RIPK1 when both glycosyltransferases were present during EPEC infection. HEK293T cells were transfected to express Flag-RIPK1$_{DD}$ and then infected with derivatives of EPEC E2348/69 for 3 hours. Flag-RIPK1$_{DD}$ was immunoprecipitated from the cell lysates, digested and subjected to LC-MS/MS. During infection with wild type EPEC we detected RIPK1 peptides containing either hexose or HexNAc modification of Arg603 (Fig 8A and S2 Table). However, Arg-GlcNAc modification was more abundant and consistently detected during wild type EPEC infection, with arginine-hexose modification of RIPK1 detected in only 3 of the 6 replicates examined (Fig 8A and S2 Table). Arg-GlcNAc modification of RIPK1 was attributable to NleB1, as this modification was not detected during infection with EPEC Δ*nleB1*, although arginine-hexose modification of Arg603 was still detected (Fig 8A and S2 Table). Likewise, arginine-hexose modification was

dependent on NleB2 during wild type EPEC infection as the modification was not detected during infection with EPEC Δ*nleB2* (Fig 8A and S2 Table). Interestingly, we observed both hexose and HexNAc modification of Flag-RIPK1$_{DD}$ when NleB2 was overexpressed in the complemented EPEC Δ*nleB1*/*nleB2* mutant, although arginine-glucose modification was more abundant (Fig 8A and S2 Table). As expected, this ratio was reversed upon complementation of the EPEC Δ*nleB1*/*nleB2* mutant with NleB2$_{S252G}$ (Fig 8A and S2 Table). Immunoblot analysis using anti-Arg-GlcNAc antibodies was consistent with the peptide analysis (Fig 8B). Arg-GlcNAc modification of Flag-RIPK1$_{DD}$ was detected by immunoblot during infection with EPEC strains that contained *nleB1* including wild type E2348/69 and Δ*nleB2* (Fig 8B). Arg-GlcNAc modification of Flag-RIPK1$_{DD}$ was also detected NleB2 was overexpressed (Fig 8B). However, the highest level of Arg-GlcNAc modification of Flag-RIPK1$_{DD}$ was observed when NleB2$_{S252G}$ was overexpressed (Fig 8B). In summary, NleB2 utilisation of UDP-GlcNAc could be detected under conditions where both the effector and host target were overexpressed, but UDP-glucose remained the preferred sugar donor of native NleB2.

## Auto-glucosylation by NleB2

Auto-GlcNAcylation is also catalysed by NleB1 and SseK1 and -3 when overexpressed [11,15,18]. To identify potential auto-glucosylated peptides in NleB2, we digested purified GST-NleB2 with Lys-C and subjected peptides to LC-MS/MS with EThcD fragmentation. We observed that GST-NleB2 isolated from BL21 *E. coli* was modified with a hexose sugar on residue Arg140 (S10A Fig), an arginine residue that is not conserved in NleB1 (S10B Fig). No Arg-GlcNAcylated peptides were detected within NleB2, suggesting that NleB2 also preferentially utilises a hexose sugar donor when expressed in bacteria in the presence of multiple nucleotide-sugars. In *E. coli*, cytoplasmic UDP-glucose levels are usually around 2.5 mM, while UDP-GlcNAc is 9.3 mM [29]. Occupancy analysis indicated an Arg-hexose occupation rate of only 0.6% (S10C Fig), thus we did not further investigate NleB2 auto-glucosylation.

## Discussion

Characterization of the activity of bacterial effector proteins has revealed new classes of enzymes as well as unique post-translational protein modifications. For example, EspL from EPEC and OspD3 from *Shigella* are members of a novel family of cysteine protease effectors that can block necroptotic cell death by cleaving and inactivating the RHIM-containing proteins, RIPK1 and RIPK3 [30,31]. NleE from EPEC and OspZ from *Shigella* are S-adenosyl-L-methionine (SAM)-dependent methyltransferases that modify a cysteine residue in the zinc finger domains of TAB2 and TAB3, thereby preventing their interaction with ubiquitinated TRAF proteins and leading to inhibition of NF-κB activation [20,32–34].

In this study, we provide further insights into the unique activity of the NleB family of arginine-glycosyltransferase effectors. Using *in vitro* assays, we found that in contrast to other NleB/SseK effectors that catalyse Arg-GlcNAc glycosylation, NleB2 from EPEC preferentially modifies arginine residues with glucose, thereby mediating an Arg-glucose linkage. This rare biochemical modification has only been detected in auto-glucosylating plant and green algae proteins [35], and has not previously been reported among bacterial or mammalian glycosyltransferases. The function of plant protein auto-Arg-glucose modification is unclear, although the modification is reversible in plants [35].

Mass spectrometry performed on the death domain of RIPK1 incubated with NleB2 *in vitro* revealed the glucose-modified residue was Arg603, the same residue in RIPK1 that is GlcNAcylated by NleB1 [9]. This arginine is conserved in many death domain proteins and is required for protein-protein interactions and subsequent signalling [4,36,37]. We also found

that, *in vitro*, NleB2 glucosylated the corresponding arginine in FADD (Arg117) and TNFR1 (Arg376) in co-transfected HEK293T cells. Although weak Arg-HexNAc modification of TRADD by NleB2 has been reported previously [4], we did not detect HexNAc nor hexose modification of TRADD when co-expressed with NleB2, similar to another study which examined *in vitro* Arg-GlcNAc modification of TRADD [18]. We also observed that, *in vitro*, Arg603 can be modified by NleB2 with two glucose moieties. The precise mechanisms by which two glucose molecules can be added to arginine are unknown, although dual methylation of arginine has been reported previously [38]. The physiological relevance of dual glucose modification of arginine is also questionable, as it was only detected in the presence of high concentrations of UDP-glucose. An unexpected finding of our *in vitro* observations was that NleB1 utilised UDP-glucose to modify RIPK1$_{DD}$ when no other sugar donors were present. The relevance of this observation during infection is unclear, as it is unlikely to occur when multiple nucleotide-sugars are present in host cell cytoplasm. Indeed, we found that when expressed at native levels during infection, only Arg-GlcNAc modification of Flag-RIPK1$_{DD}$ could be detected by NleB1, whereas arginine-hexose was the dominant modification mediated by NleB2.

We identified the amino acid residues Ser252 in NleB2 and Gly255 in NleB1 that play significant roles in the nucleotide-sugar preference of each enzyme. Although extensive mutational analysis and structural characterisation of NleB1 has been performed [9,13,18], mutation of Gly255 had not previously been examined. Interestingly, this residue is located immediately downstream of Glu253 and Asn254 of the HEN motif which was identified by Park *et al.* to be essential for NleB1 and SseK catalytic activity [18]. We and others have also shown that Glu253 is essential for NleB1 Arg-GlcNAc activity [4,9,13]. The region containing Gly255 is disordered in the published structure of NleB1 in complex with FADD and UDP [9]. However, in the published structure of SseK2 in complex with UDP-GlcNAc [18], the corresponding residue (Gly273) was located near the *N*-acetyl group of the GlcNAc moiety. Although this residue has not been shown to interact with UDP-GlcNAc directly, structural modelling of SseK3 with UDP and GlcNAc suggested that the conserved glycine (Gly260) forms a hydrogen bond with the GlcNAc *N*-acetyl group [39]. We propose that a serine instead of glycine in this position in NleB2, may reduce the size of the binding-pocket that accommodates the *N*-acetyl group of UDP-GlcNAc, resulting in a preference for the smaller UDP-glucose. Studies of β1, 4-Galactosyltransferase I (β4Gal-T1) show a similar phenomenon [24]. Gal-T1 normally utilises UDP-galactose, but also has a very low GalNAc-transferase activity [24]. Resolution of the structure of Gal-T1 revealed that a tyrosine residue formed a hydrogen bond with the *N*-acetyl group of the GalNAc moiety, and that when this residue was mutated to leucine, isoleucine or asparagine, this bond was disrupted and GalNAc-transferase activity was increased [24]. Likewise, we observed that mutation of Ser252 in NleB2 to glycine increased GlcNAc-transferase activity, suggesting that this residue may interact with the *N*-acetyl group of the GlcNAc moiety. The reciprocal G255S mutation in NleB1 resulted in a change in the nucleotide-sugar utilised by NleB1 from UDP-GlcNAc to UDP-glucose. We found that although only weak Arg-GlcNAcylation was detected in the presence of NleB1$_{G255S}$ during infection or when expressed in mammalian cells, NleB1$_{G255S}$ nonetheless inhibited inflammatory and cell death signalling similar to NleB1. Thus, we conclude that Arg-glucosylation inhibits protein-protein interactions and downstream signalling similar to Arg-GlcNAcylation.

Small changes in amino acid sequence are known to change glycosyltransferase nucleotide-sugar usage. For example, two glycosyltransferases from *Pasteurella multocida*, GctD and GatB, are identical except for one amino acid residue, resulting in differences in sugar donor usage from UDP-glucose to UDP-galactose respectively, with a consequent impact on LPS

structure [22]. Point mutations can also be designed to engineer glycosyltransferases that utilise non-canonical sugar donors, creating highly useful research tools [40]. The significance of NleB2 as a glucosyltransferase rather than GlcNAc transferase is unclear. Cytoplasmic concentrations of UDP-GlcNAc can fluctuate, including during the immune response [41]. This has been shown to have significant effects on host cellular enzymes including the O-GlcNAc transferase, OGT [41]. The utilization of UDP-glucose by NleB2 may allow EPEC to adapt to low availability of cytoplasmic UDP-GlcNAc during infection, meaning NleB2 could inhibit signalling when the Arg-GlcNAc activity of NleB1 is reduced.

We observed both HexNAc and hexose modification of Arg603 on ectopically expressed Flag-RIPK1$_{DD}$ during EPEC infection. However, the lack of antibody-based tools to immunoprecipitate arginine-hexose modifications in a non-biased way, means we could not investigate NleB2 activity against endogenous target proteins during EPEC infection. Analysis of endogenous RIPK1 is further complicated by our previous observations that RIPK1 is cleaved by EspL during EPEC infection [30]. This may explain, in part, why there is a lack of phenotype associated with the EPEC E2348/69 Δ*nleB2* mutant in the assays utilised to date. Analysis is further confounded by other effectors such as NleE, NleC and NleB1 which also target RIPK1-dependent signalling pathways [4,5,20,32,33,42–46]. Analysis of effector translocation by EPEC E2348/69 has not revealed significant differences between NleB1 and NleB2, which are both translocated into host cells at low levels compared to other effectors such as NleA [47]. However, NleB1 and NleB2 are located on different genomic islands (IE6 and PP4 respectively), and there is some evidence that NleB2 expression may be regulated independently from NleB1 [48]. Future investigation of NleB2 translocation and activity under different metabolic conditions during EPEC infection may reveal the function of NleB2 and purpose of utilising UDP-glucose over UDP-GlcNAc.

Given the complex interaction of EPEC with host cells during infection, it is possible that EPEC modulates host metabolism, as reported for other pathogens [49]. Intriguingly, NleB1 also directly influences host glucose metabolism via Arg-GlcNAc modification of HIF-1α, although it is not clear whether this occurs at endogenous levels of effector translocation [50]. The preference of NleB2 for UDP-glucose may help overcome metabolic limitations that occur during EPEC infection. Although our previous work suggested that NleB2 does not have a major impact on inflammatory or cell death signalling during EPEC infection [5], it is possible that NleB2 glucosylates non-death domain targets, and future work will require novel tools to identify the Arg-glucose modified targets of NleB2 during infection.

## Materials and methods

### Bacterial strains and growth conditions

The bacterial strains used in this study are listed in Table 1. Bacteria were grown in Luria-Bertani (LB) broth or Roswell Park Memorial Institute medium (RPMI) with GlutaMAX (Gibco) at 37˚C in the presence of ampicillin (100 μg/ml), kanamycin (100 μg/ml) or chloramphenicol (25 μg/ml) when required.

### DNA cloning and purification

The plasmids used in this study are listed in Table 1. Primers used in this study are listed in Table 2. DNA modifying enzymes were used in accordance with manufacturer instructions (Roche). Plasmids were extracted using the QIAGEN QIAprep Spin Miniprep Kit or the QIAGEN Plasmid Midi Kit. PCR products and restriction digests were purified using the Wizard SV Gel and PCR Clean-Up System (Promega). pGBKT7-NleB2 was generated by amplifying the *nleB2* gene from EPEC E2348/69 genomic DNA using primer pair NleB2$_{F1}$/NleB2$_{R1}$ before

**Table 1. Bacterial strains and plasmids used in this study.**

| Strain/plasmid | Characteristics | Source/Reference |
|---|---|---|
| EPEC E2348/69 | Wild type EPEC O127:H6 | [59] |
| Δ*nleB1* | EPEC E2348/69 Δ*nleB1* | [5] |
| Δ*nleB2* | EPEC E2348/69 Δ*nleB2* | [5] |
| Δ*nleB1/nleB2* | EPEC E2348/69 Δ*nleB1* and Δ*nleB2* | [5] |
| BL21 C43 (DE3) | *E. coli* used for expression of proteins for affinity purification | Novagen |
| *Saccharomyces cerevisiae* PJ69-4A | Yeast strain for performing yeast-2-hybrid interaction assays. (MATa, trp1-901, leu2-3,112,ura3-52 his3-200,gal4Δ, gal80Δ,LYS2::GAL1-HIS3, GAL2-ADE2, met2::GAL7-lacZ) | Clontech |
| pGBKT7 | Yeast expression vector for fusion of GAL4 DNA binding domain to protein of interest. Carries *TRP1* nutritional marker for selection in yeast. | Clontech |
| pGBKT7-NleB1 | *nleB1* from EPEC E2348/69 in pGBKT7 | This study |
| pGBKT7-NleB2 | *nleB2* from EPEC E2348/69 in pGBKT7 | This study |
| pGADT7 AD | Yeast expression vector for fusion of GAL4 activation domain to protein of interest. Carries *LEU2* nutritional marker for selection in yeast. | Clontech |
| pGADT7-RIPK1$_{DD}$ | Death domain (residues 590–675) of RIPK1 in pGADT7 AD | This study |
| pGADT7-TNFR1$_{DD}$ | Death domain (residues 358–438) of TNFR1 in pGADT7 AD | This study |
| pTrc99A | Cloning vector for expression of proteins from P*trc* | Pharmacia Biotech |
| pNleB1 | *nleB1* from EPEC E2348/69 in pTrc99A | [20] |
| pNleB1$_{G255S}$ | *nleB1* from EPEC E2348/69 carrying mutation G255S in pTrc99A | This study |
| pEGFP-C2 | N-terminal Green fluorescent protein (GFP) expression vector | Clontech |
| pGFP-NleB1 | *nleB1* from EPEC E2348/69 in pEGFP-C2 | [5] |
| pGFP-NleB1$_{DXD}$ | *nleB1* from EPEC E2348/69 carrying alanine substitutions for residues D$_{221}$AD in pEGFP-C2 | [5] |
| pGFP-NleB1$_{G255S}$ | *nleB1* from EPEC E2348/69 carrying mutation G255S in pEGFP-C2 | This study |
| pGFP-NleB2 | *nleB2* from EPEC E2348/69 in pEGFP-C2 | [5] |
| pGFP-NleB2$_{DXD}$ | *nleB2* from EPEC E2348/69 carrying alanine substitutions for residues D$_{218}$MD in pEGFP-C2 | This study |
| pGFP-NleB2$_{S252G}$ | *nleB2* from EPEC E2348/69 carrying mutation S252G in pEGFP-C2 | This study |
| p3xFlag-*Myc*-CMV-24 | Dual tagged N-terminal Met-3xFlag and C-terminal *c-myc* expression vector | Sigma-Aldrich |
| pFlag-TRADD | Human TRADD in p3xFlag-*Myc*-CMV | Jürg Tschopp |
| pFlag-TNFR1$_{DD}$ | Death domain of human TNFR1 in p3xFlag-Myc-CMV-24 | [11] |
| pFlag-RIPK1$_{DD}$ | Death domain of human RIPK1 in p3xFlag-Myc-CMV-24 | Thus study |
| pRL-TK | Renilla luciferase vector | Promega |
| pNF-κB-Luc | Vector for measuring NF-κB dependent luciferase expression | Clontech |
| pGEX-4T-1 | N-terminal glutathione S-transferase (GST) cloning/expression vector | GE Healthcare |
| pGEX-NleB1 | *nleB1* from EPEC E2348/69 in pGEX-4T-1 | [5] |
| pGEX-NleB1$_{G255S}$ | *nleB1* from EPEC E2348/69 carrying the mutation G255S in pGEX-4T-1 | This study |
| pGEX-NleB1$_{DXD}$ | *nleB1* from EPEC E2348/69 carrying the mutation D$_{221}$AD in pGEX-4T-1 | [5] |
| pGEX-NleB2 | *nleB2* from EPEC E2348/69 in pGEX-4T-1 | This study |
| pGEX-NleB2$_{DXD}$ | *nleB2* from EPEC E2348/69 carrying alanine substitutions for residues D$_{218}$MD in pGEX-4T-1 | This study |
| pGEX-NleB2$_{S252G}$ | *nleB2* from EPEC E2348/69 carrying the mutation S252G in pGEX-4T-1 | This study |
| pMAL-c2X | N-terminal maltose-binding protein (MBP) Tag cloning/expression vector | New England Biolabs |
| pMAL-RIPK1$_{DD}$ | Death domain (residues 583–668) of RIPK1 in pMAL-c2X | This study |
| pMAL-RIPK1$_{DD(R603A)}$ | Death domain (residues 583–668) of RIPK1 carrying the mutation R603A in pMAL-c2X | This study |
| pHis-FADD | FADD in pET28a in frame with N-terminal 6xHis tag for affinity purification | [5] |
| pHis-FAS$_{DD}$ | Death domain (residues 223–319) of human FAS in pET28a in frame with N-terminal 6xHis tag for affinity purification | This study |
| pHis-TRAILR2$_{DD}$ | Death domain (residues 333–428) of human of human TRAILR2 in pET28a in frame with N-terminal 6xHis tag for affinity purification | [11] |

**Table 2. List of primers used in this study.**

| Name | Primer sequences 5'-3' |
|---|---|
| NleB2$_{F1}$ | CGGAATTCATGCTTTCACCGATAAGGACAACTTTC |
| NleB2$_{R1}$ | CGGGATCCTTACCATGAACTGCATGTATACTGAC |
| NleB2$_{F2}$ | CGGGATCCATGCTTTCACCGATAAGGACAACTTTC |
| NleB2$_{R2}$ | CGGAATTCTTACCATGAACTGCATGTATACTGAC |
| RIPK1$_{F}$ | CGCGAATTCATGACGGATAAACACCTGGACCC |
| RIPK1$_{R}$ | CGCGTCGACTTAGACGTAAATCAAGCTGCTCAG |
| RIPK1$_{F2}$ | CGAAGCTTACGGATAAACACCTGGACC |
| RIPK1$_{R2}$ | CGGAATTCCTAGACGTAAATCAAGCTGCTC |
| FAS$_{F}$ | CGCGAATTCATGAATTTATCTGATGTTGACTTGAG |
| FAS$_{R}$ | CGCGAATTCCTAAGTAATGTCCTTGAGGATGATAG |
| TNFR1$_{F}$ | CGCCATATGATGACGCTGTACGCCGTGGTGG |
| TNFR1$_{R}$ | CGCGGATCCTCACTCGATGTCCTCCAGGCAGC |
| NleB2$_{(DXD)F}$ | GAGGGGTGTATCTATCTTGCTGCGGCTATGATACTTACAGGTAAGC |
| NleB2$_{(DXD)R}$ | GCTTACCTGTAAGTATCATAGCCGCAGCAAGATAGATACACCCCTC |
| NleB2$_{(S252G)F}$ | CGTCGTAATGATAGTGTAAATATTGAAAATGGTGCAATAATTGTTAACCG |
| NleB2$_{(S252G)R}$ | CGGTTAACAATTATTGCACCATTTTCAATATTTACACTATCATTACGACG |
| NleB1$_{(G255S)F}$ | GATGAGATAAAAAGTCTTGAAAATAGTGCGATAGTTGTCAATC |
| NleB1$_{(G255S)R}$ | GATTGACAACTATCGCACTATTTTCAAGACTTTTTATCTCATC |
| RIPK1$_{(R603A)F}$ | GCACTGGAAAAACTGTGCCGCTAAACTGGGCTTCACAC |
| RIPK1$_{(R603A)R}$ | GTGTGAAGCCCAGTTTAGCGGCACAGTTTTTCCAGTGC |

the PCR product was digested with EcoRI/BamHI and ligated into pGBKT7. pGBKT7-NleB1 was constructed by digesting pGBT9-NleB1 [5], which carries *nleB1* flanked between the restriction sites EcoRI and BamHI, and ligating into pGBKT7 digested with EcoRI/BamHI. pGADT7-RIPK1$_{DD}$ was constructed by amplifying the death domain of *RIPK1* from HeLa cDNA using primer pair RIPK1$_{F}$/RIPK1$_{R}$, before being digested with EcoRI/SacI and ligated into pGADT7 AD. pGADT7-TNFR1$_{DD}$ was constructed by amplifying the death domain of *TNFR1* using primer pair and TNFR1$_{F}$/TNFR1$_{R}$ before being digested with NdeI/BamHI and ligated into pGADT7 AD. pGEX-NleB2 was generated by amplifying the *nleB2* gene from EPEC E2348/69 genomic DNA using primer pair NleB2$_{F2}$/NleB2$_{R2}$ before the PCR product was digested with EcoRI/BamHI and ligated into pGEX4T-1. pMAL-RIPK$_{DD}$ was generated by amplifying the death domain of *RIPK1* from HeLa cDNA using primer pair RIPK1$_{F}$/RIPK1$_{R}$ before the PCR product was digested with EcoRI/SacI and ligated into pMAL-c2X. pHis-FAS$_{DD}$ was generated by amplifying the death domain of *FAS* from HeLa cDNA using primer pair FAS$_{F}$/FAS$_{R}$ before the PCR product was digested with EcoRI and ligated into pET28a. pFlag-RIPK$_{DD}$ was generated by amplifying the death domain of *RIPK1* from HeLa cDNA using primer pair RIPK1$_{F2}$/RIPK1$_{R2}$ before the PCR product was digested with HindIII/EcoRI and ligated into pMAL-c2X.

## Site-directed mutagenesis

pGFP-NleB2$_{DXD}$ and pGEX-NleB2$_{DXD}$ were generated using the Stratagene QuikChange II Site-Directed Mutagenesis Kit according to manufacturer's protocol, using primer pair NleB2$_{(DXD)F}$/NleB2$_{(DXD)R}$ and pGFP-NleB2 or pGEX-NleB2 as template DNA respectively. pGFP-NleB2$_{S252G}$ and pGEX-NleB2$_{S252G}$ were generated using primer pair NleB2$_{(S252G)F}$/NleB2$_{(S252G)R}$ and pGFP-NleB2 or pGEX-NleB2 as template DNA respectively. pGFP-NleB1$_{G255S}$, pGEX-NleB1$_{G255S}$ and pNleB1$_{G255S}$ were generated using primer pair

NleB1$_{(G255S)F}$/NleB1$_{(G255S)R}$ and either pGFP-NleB1, pGEX-NleB1 or pNleB1 as template DNA respectively. pMAL-RIPK1$_{DD(R603A)}$ was generated using primer pair RIPK1$_{(R603A)F}$/RIPK1$_{(R603A)R}$ using pMAL-RIPK1$_{DD}$ as template. Plasmids were digested with DpnI at 37˚C overnight before subsequent transformation into XL1-Blue cells.

## Yeast-2-hybrid co-transformation

The yeast strain *Saccharomyces cerevisiae* PJ69-4A was co-transformed with derivatives of pGBKT7 or pGADT7 AD using the lithium acetate method [51]. Briefly, *S. cerevisiae* PJ69-4A was streaked on YPD plates supplemented with adenine and incubated at 30˚C for 3 days. Two or three pink colonies from the streak plates were then used to inoculate a 10 ml YPD broth supplemented with adenine and grown at 30˚C overnight at 200 rpm. The overnight culture was subinoculated into fresh YPD broth containing adenine at a starting OD$_{600}$ of 0.20 and grown at 30˚C to an OD$_{600}$ of 0.6–0.8. The yeast culture was then centrifuged at 4,000 rpm for 7 min and the pellet was resuspended in distilled water before being spun down again. The yeast pellet was then resuspended in 100 mM lithium acetate and centrifuged. The lithium acetate was removed and the cells were resuspended in 400 mM lithium acetate, vortexed and centrifuged. The lithium acetate was removed from the yeast pellet and polyethylene glycol (PEG 3350; Sigma-Aldrich) was added, followed by 1 M lithium acetate, herring sperm DNA (ssDNA) at a final concentration of 2 mg/ml, water and the plasmid DNA of interest before mixing by vortex. The mixture was then incubated 30˚C for 30 mins and heat shocked at 42˚C for 20 mins before being pelleted. The transformation mixture was then removed and the pellet was resuspended in distilled water and plated on selective yeast plates which were incubated at 30˚C for 3 days. Agar plates lacking leucine and tryptophan were used to select for plasmid maintenance, and agar plates lacking leucine, tryptophan, histidine and adenine were used to select for yeast harbouring interacting proteins.

## Mammalian cell culture and transfection

HeLa and HEK293T cells were cultured in in T75 cm$^2$ tissue culture flasks (Corning) in Dulbecco's modified Eagle's medium (DMEM) GlutaMAX (Gibco) supplemented with 10% Foetal Bovine Serum (FBS Bovogen Biologicals) in 5% CO$_2$ at 37˚C. Approximately 24 h before transfection, HeLa or HEK293T cells were seeded into 24 well tissue culture trays (Greiner Bio-One) at a density of 10$^5$ cells per well. Cells were transfected with pEGFP-C2 derivatives using FuGENE 6 Transfection Reagent (Promega) according to manufacturer instructions. Cells were transfected for 24 h before being lysed for immunoblot analysis.

## NF-κB-dependent luciferase reporter assay

HeLa cells were seeded onto 24 well trays and co-transfected with derivatives of pEGFP-C2 (0.4 μg) and 0.2 μg of pNF-κB-Luc (Clontech, Palo Alto CA, USA) and 0.05 μg of pRL-TK (Promega, Madison WI, USA). After 24 hours of transfection, cells were either left untreated, or stimulated with TNF (20 ng/ml) and incubated in 5% CO$_2$ at 37˚C for a further 16 hours. A dual-luciferase reporter assay was then performed on the HeLa cell lysates according to the manufacturer's protocols (Promega Part# TM040). Samples were measured on a CLARIOstar microplate reader (BMG Labtech).

## EPEC infection of HT-29 cells

HT-29 cells were cultured in T75 cm$^2$ tissue culture flasks (Corning) in RPMI GlutaMAX (Gibco) supplemented with 10% Foetal Bovine Serum (FBS Bovogen Biologicals) in 5% CO$_2$ at

37˚C. Two days before infection HT-29 cells were seeded into 24 well tissue culture trays at a density of $2x10^5$ cells per well. The day before infection derivatives of EPEC were inoculated into LB broth and grown with shaking at 37˚C overnight. On the day of infection, overnight cultures of EPEC were sub-cultured 1:75 in RPMI GlutaMAX (Gibco) and grown statically for 3 h at 37˚C with 5% CO2. Where necessary, cells were induced with 1 mM IPTG (Sigma) for 30 min before infection. HT-29 cells were washed twice with PBS and infected with EPEC grown to an $OD_{600}$ of 0.06 for 3 h before being lysed for immunoblot analysis. Where required, after 3 hours of EPEC infection the HT-29 cells were treated for a further 2 hours with 100 μg/ml gentamycin to stop the infection, in combination with 20 ng/ml FasL before being lysed for immunoblot analysis.

## Immunoprecipitation of Flag-tagged fusion proteins

For immunoprecipitation of Flag-TNFR1$_{DD}$ and Flag-TRADD from co-transfected cells, $10^7$ HEK293T cells were seeded into 15 cm dishes (Corning) the day before transfection. Cells were co-transfected with either pFlag-TNFR1$_{DD}$ or pFlag-TRADD and either pGFP-NleB2 or pGFP-NleB2$_{DXD}$ at a ratio of 2:1 of Flag:GFP plasmid DNA using FuGENE 6 Transfection Reagent (Promega) according to manufacturer instructions. 16 hours after transfection, cells were washed once with PBS before being lysed in ice cold lysis buffer (50 mM Tris-HCl pH 7.4, 150 mM NaCl, 1 mM EDTA, 1% Triton x-100, 10 mM NaF, 1 mM PMSF, 2 mM $Na_3VO_4$ with 1 x EDTA-free Complete Protease Inhibitor Cocktail (Roche)) for 30 minutes. Cell debris was pelleted and the supernatant added to equilibrated anti-Flag magnetic beads (Sigma-Aldrich) and incubated at 4˚C for 2 hours. Beads were washed with lysis buffer before bound proteins were eluted in MilliQ containing 5% SDS by heating at 95˚C for 10 minutes. Beads were separated from buffers using a magnetic separation rack (New England Biolabs). Eluates were subjected to S-trap based protein clean up and mass spec analysis (see below).

For immunoprecipitation of Flag-RIPK1$_{DD}$ from EPEC-infected cells, $10^7$ HEK293T cells were seeded into 15 cm dishes 2 days before infection. The day before infection HEK293T cells were transfected with pFlag-RIPK$_{DD}$ using FuGENE 6 Transfection Reagent (Promega) according to manufacturer instructions and derivatives of EPEC were inoculated into LB broth and grown with shaking at 37˚C overnight. 16 hours after transfection, the HEK293T cells were washed with PBS, and media was replaced to serum-free DMEM GlutaMAX (Gibco). Overnight EPEC cultures were sub-cultured 1:75 in DMEM GlutaMAX (Gibco) and grown statically for 3 h at 37˚C with 5% CO2. Where necessary, cells were induced with 1 mM IPTG (Sigma) for 30 min before infection. HEK293T cells were infected with 25 mL of EPEC cultures at an $OD_{600}$ of 0.01 for 3 hours before being washed with PBS and lysed for Flag-immunoprecipitation as above.

## Immunoblot analysis

Transfected or infected mammalian cells were lysed in cold lysis buffer (50 mM Tris-HCl pH 7.4, 150 mM NaCl, 1 mM EDTA, 1% Triton x-100, 10 mM NaF, 1 mM PMSF, 2 mM $Na_3VO_4$ with 1 x EDTA-free Complete Protease Inhibitor Cocktail (Roche)). Cell debris was pelleted and the supernatant added to 4×Bolt LDS Sample Buffer (Thermo Fisher) with 50 mM DTT before being heated to 70˚C for 10 min.

Protein samples were resolved on Bolt 4–12% Bis-Tris Plus Gels (Thermo Fisher) by PAGE. Proteins were then transferred to nitrocellulose membranes using the iBlot2 Dry Blotting system (Thermo Fisher) before the nitrocellulose membranes were blocked for 1 h with 5% skim milk powder in Tris buffered saline (TBS; 50mM Tris-HCl pH 7.5, 150mM NaCl) with 0.1% Tween 20 (Biochemicals). Nitrocellulose membranes were washed with TBS containing 0.1%

Tween 20 and probed overnight at 4˚C with one of the following primary antibodies diluted 1:1000 in TBS with 0.1% Tween 20 and 5% bovine serum albumin (BSA, Sigma): rabbit monoclonal anti-Arginine (GlcNAc) (EPR18251; Abcam), mouse anti-GFP (7.1 and 13.1; Roche), mouse monoclonal anti-β-actin (AC-15; Sigma), mouse monoclonal anti-caspase 8 (1C12; Cell Signaling), mouse monoclonal anti-MBP (New England Biolabs), mouse monoclonal anti-histidine tag (AD1.1.10; Bio-Rad), or rabbit polyclonal anti-GST (Cell Signaling). Rabbit polyclonal anti-NleB1 (made by Chemicon International against antigen GST-NleB1) was diluted 1:100 in TBS with 0.1% Tween 20 and 5% BSA and incubated overnight at 4˚C. Secondary antibodies were horseradish peroxidase (HRP)-labelled anti-mouse IgG (PerkinElmer) or HRP-labelled anti-rabbit IgG (PerkinElmer), which were diluted 1:3000 in TBS with 0.1% Tween 20 and 5% BSA and incubated on nitrocellulose membranes for 1 hour at room temperature. Immunoblots were routinely developed with ECL Western Blotting Detection Regents (Cytiva), or when a stronger signal was required the immunoblots were developed with Western Lightning Ultra (PerkinElmer). Images were acquired with the Amersham Imager 680 (Cytiva).

## Protein purification

Plasmids for the expression of MBP-tagged RIPK1$_{DD}$, 6xHis-tagged FADD, FAS$_{DD}$ or TRAILR2$_{DD}$ or GST-tagged NleB1, NleB2 or derivatives were transformed into BL21 C43 (DE3) E. coli. LB overnight cultures of BL21 containing the appropriate expression vector were used to inoculate a 200 ml LB broth 1:100 which was grown at 37˚C with shaking to an optical density (A600) of 0.6. Cultures were induced with 1 mM IPTG and grown for a further 16 h at 18˚C before being pelleted by centrifugation. Before purification, bacterial pellets were resuspended in the appropriate binding buffer from Novagen His-Bind and GST-Bind kits, or in TBS for MBP- RIPK1$_{DD}$. Bacterial suspensions were lysed using the EmulsiFlex-C3 High Pressure Homogenizer (Avestin) according to manufacturer's instructions. Purification of proteins was performed according to manufacturer's protocols (Novagen His-Bind and GST-Bind kits and NEB Amylose resin). Proteins were then desalted using PD-10 desalting columns (GE) and protein concentrations were determined using a bicinchoninic acid (BCA) kit (Thermo Scientific).

## UDP-Glo assay

Vmax and Km calculations were performed using data from UDP-Glo assays (Promega) similar to previous studies [28,52]. Purified GST, GST-tagged NleB1, NleB1$_{G255S}$, NleB1$_{DXD}$, NleB2, NleB1$_{S252G}$ or NleB2$_{DXD}$ at 150 nM concentrations were incubated with 1 µM MBP-RIPK1$_{DD}$ in 1x glycosyltransferase buffer (50 mM Tris-HCL pH 7.4, 100 mM NaCl, 10 mM MgCl$_2$, 1 mM MnCl$_2$) with titrated UDP-GlcNAc or UDP-glucose (0–1 mM) for 30 minutes at 30˚C. The UDP Glo assay was then performed according to manufacturer instructions, and luminescence was measured using a CLARIOstar microplate reader (BMG Labtech). Readings obtained from GST incubated with MBP-RIPK1$_{DD}$ in the presence of UDP sugar donors were subtracted from all other readings as background luminescence. Relative light units (RLU) measured at the end of the reaction were divided by 30 to calculate RLU/min produced, and Vmax and Km values were calculated by using the non-linear regression fit Michaelis-Menten equation in GraphPad Prism Version 6.

## *In vitro* glycosylation assays

For intact mass spectrometry analysis, 10 µg of purified MBP-RIPK1$_{DD}$ was incubated with 1 µg of purified GST-NleB1, GST-NleB2 or mutants for 3 hours at 37˚C in TBS supplemented

with 10 mM $MgCl_2$ and 10 mM $MnCl_2$ in the presence of sugar donors. UDP-GlcNAc, UDP-glucose, UDP-GalNAc, UDP-galactose, UDP-GlcA or GDP-mannose (Sigma) were used at 10 mM, 50 μM or 0.5 μM where indicated and when incubated individually, or at 25 μM concentrations when co-incubated. For immunoblot analysis, 3 μg of purified MBP-RIPK1$_{DD}$ or His-FADD were incubated with 1 μg of purified GST-NleB1, GST-NleB2 or mutants for 3 hours at 37°C in TBS supplemented with 10 mM $MgCl_2$ and 10 mM $MnCl_2$ in the presence of sugar donors at 25 μM.

## Intact protein-based MS analysis

Intact analysis was performed using either a 6220 or a 6520 Accurate mass Q-TOF mass spectrometer (Agilent). Protein samples were re-suspended in 2% acetonitrile, 0.1% TFA and 2–10 μg of in vitro glycosylated protein injected onto a C5 Jupiter column (5 μm, 300 Å, 50 mm × 2.1 mm, Phenomenex) using an Agilent 1200 series HPLC system. Samples were desalted by washing the column with buffer A (2% acetonitrile, 0.1% formic acid) for 4 minutes and then separated with a 12 minutes linear gradient from 2 to 100% buffer B (80% acetonitrile, 0.1% formic acid) at a flow rate of 0.250 ml/min prior to reconditioning of the column for 4 minutes. MS1 mass spectra were acquired at 1 Hz between a mass range of 300–3,000 *m/z*. Intact mass analysis and deconvolution was performed using MassHunter B.06.00 (Agilent).

## Peptide-based MS analysis

For peptide analysis, 10 μg of purified MBP-RIPK1$_{DD}$ was incubated alone, or with 1 μg of purified GST-NleB2 for 3 hours at 37°C in TBS supplemented with 10 mM $MgCl_2$ and 10 mM $MnCl_2$ in the presence of either 10 mM UDP-GlcNAc, UDP-glucose or UDP-galactose. 10 μg of purified His-FADD, His-FAS$_{DD}$ or His-TRAIL2$_{DD}$ were incubated alone, or with 1 μg of purified GST-NleB2 as above in the presence of either 10 mM or 50 μM UDP-glucose.

## TFE based sample preparation of MBP-RIPK1$_{DD}$/ MBP-RIPK1$_{R603A}$ and purified NleB2

*In vitro* glycosylated MBP-RIPK1$_{DD}$/ MBP-RIPK1$_{R603A}$ and purified NleB2 samples were resuspend in 50ul 20% TFE and diluted equal volume of reduction/alkylation buffer (40mM TCEP, 80mM chloroacetamide and 100mM $NH_4HCO_3$). Samples were then heated at 40°C for 30 min to aid denaturation and reduction / alkylation in the dark. Glu-C (Promega) for RIPK1 or Lys-C (Wako Chemicals) for NleB2 was added (1/50 w/w) and allowed to incubate overnight at 37°C. Digested samples were acidified to a final concentration of 0.5% formic acid and desalted using C18 stage tips [53] before analysis by LC-MS.

## S-trap based sample preparation of In-vitro glycosylated 6xHis-tagged FADD, FAS$_{DD}$ or TRAILR2$_{DD}$; immunoprecipitated Flag-TNFR1$_{DD}$ and Flag-TRADD and immunoprecipitated RIPK1-DD from infection samples

In-vitro glycosylated 6xHis-tagged FADD, FAS$_{DD}$ or TRAILR2$_{DD}$; immunoprecipitated Flag-TNFR1$_{DD}$ and Flag-TRADD and immunoprecipitated RIPK1-DD from infection samples were prepared using S-trap columns (Protifi, USA) according to the manufacturer's instructions. Briefly samples were adjusted to contain 5% SDS, boiled for 10 minutes with 10mM DTT, allowed to cool to room temperature then alkylated with 40mM of iodoacetamide for 30 minutes in the dark. Samples were then acidified with phosphoric acid to a final concentration of 1.2% and mixed with seven volumes of 90% methanol/100mM TEAB pH 7.1 before being applied to S-trap mini columns. Samples were washed four times with 90%

methanol/100mM TEAB pH 7.1 to remove SDS then 2μg of trypsin (Promega, USA) in 100mM TEAB pH8.5 added to 6xHis-tagged FADD, $FAS_{DD}$ or $TRAILR2_{DD}$; Flag-$TNFR1_{DD}$ and Flag-TRADD samples while 1.25μg of GluC (Promega, USA) in 100mM ammonium bicarbonate was added to immunoprecipitated RIPK1 samples. Proteases were spun through the S-trap columns and samples digested for 4 hours at 47˚C for trypsin or 37˚C for GluC digests. Peptides were collected from the S-traps by washing with 100mM TEAB pH8.5 or 100mM ammonium bicarbonate followed by 0.2% Formic acid followed by 0.2% Formic acid/ 50% acetonitrile. Peptide washes were pooled, dried and then resuspended in Buffer A* (0.1% TFA, 2% acetonitrile) before being cleaned up with home-made StageTips composed of 1 mg Empore™ C18 material (3M) and 1 mg of OLIGO R3 reverse phase resin (Thermo Fisher Scientific, USA) as previously described [53,54]. Columns were wet with Buffer B (0.1% formic acid, 80% acetonitrile) and conditioned with Buffer A* prior to use. Resuspended samples were loaded onto conditioned columns, washed with 10 bed volumes of Buffer A* and bound peptides were eluted with Buffer B before being dried then stored at -20˚C.

## LC-MS analysis of *In-vitro* glycosylated RIPK1 and NleB2

Peptide samples were re-suspended in Buffer A* and separated using a two-column chromatography set up composed of a PepMap100 C18 20 mm x 75 μm trap and a PepMap C18 500 mm x 75 μm analytical column (Thermo Fisher Scientific). Samples were concentrated onto the trap column at 5 μL/min for 5 minutes with Buffer A (0.1% formic acid, 2% DMSO) then infused into an Orbitrap Fusion™ Lumos™ Tribrid™ or a Q-Exactive plus Mass Spectrometer (Thermo Fisher Scientific) at 300 nL/minute via the analytical column using a Dionex Ultimate 3000 UHPLC (Thermo Fisher Scientific). 95-minute analytical runs were undertaken by altering the buffer composition from 2% Buffer B (0.1% formic acid, 77.9% acetonitrile, 2% DMSO) to 28% B over 60 minutes, then from 28% B to 40% B over 10 minutes, then from 40% B to 100% B over 2 minutes. The composition was held at 100% B for 3 minutes, and then dropped to 2% B over 5 minutes before being held at 2% B for another 15 minutes. The Orbitrap Fusion Lumos Tribrid Mass Spectrometer was operated in a data-dependent mode, automatically switching between the acquisition of a Orbitrap MS scan (120,000 resolution) every 3 seconds and the fragmentation of precursors with either EThcD (NCE 15%, maximal injection time of 200 ms with an AGC of 2e5) or stepped collision energy HCD scan (using NCE 30%, 35%, 45% with a maximal injection time of 200 ms and a AGC of 2e5) analysed within the Orbitrap mass analyzer (resolution of 30k or 15k). The Q-Exactive plus Mass Spectrometer was operated in a data-dependent mode, acquiring one full precursor scan (resolution 70,000; 440–2000 *m/z*, AGC target of $1\times10^6$) followed by 10 data-dependent HCD MS-MS events (resolution 35k AGC target of $2\times10^5$ with a maximum injection time of 200 ms).

## LC-MS analysis of *In-vitro* glycosylated 6xHis-tagged FADD, $FAS_{DD}$ or $TRAILR2_{DD}$; immunoprecipitated Flag-$TNFR1_{DD}$ and Flag-TRADD and immunoprecipitated RIPK1-DD from infection samples

Stagetip cleaned up samples were re-suspended in Buffer A* and separated using a two-column chromatography set up composed of a PepMap100 C18 20 mm x 75 μm trap and a PepMap C18 500 mm x 75 μm analytical column (Thermo Fisher Scientific) coupled to a Orbitrap Fusion™ Eclipse™ or Exploris™ 480 Mass Spectrometer (Thermo Fisher Scientific) for immunoprecipitated proteins or an Orbitrap Q-Exactive plus Mass Spectrometer (Thermo Fisher Scientific) for in vitro glycosylated samples. Samples were infused at 300 nl/minute via analytical columns using Dionex Ultimate 3000 UPLCs (Thermo Fisher Scientific) on all systems. 95-minute gradients were run for each sample altering the buffer composition from 2% Buffer

B to 23% B over 65 minutes, then from 23% B to 40% B over 20 minutes, then from 40% B to 80% B over 4 minutes, the composition was held at 80% B for 2 minutes, and then dropped to 2% B over 0.1 minutes and held at 2% B for another 4.9 minutes. The Eclipse™ and Exploris™ Mass Spectrometers were operated in a hybrid data-dependent and data-independent mode collecting 2.0 seconds of data-dependent scans followed by 1.4 seconds of data-independent scans. For data-dependent scans a single Orbitrap MS scan (300–1600 m/z, maximal injection time of 25 ms, an AGC of 300% and a resolution of 120k) was acquired followed by Orbitrap MS/MS HCD scans of precursors (NCE 30%, maximal injection time of 40 ms, an AGC of 200% and a resolution of 15k). After each round of data-dependent scans data-independent scans targeting the +3, +4 and +5 charge states of the RIPK1-DD peptide NLGKHWKNC ARKLGFTQSQIDE in its unmodified, HexNAc modified and Hex modified states were undertaken (corresponding to the m/zs: 877.4467; 658.3369; 526.8709; 931.4643; 698.8501; 559.2815; 945.1398; 709.1067 and 567.4868). Each m/z were isolated and fragmented using stepped collision energy HCD scans (using the NCE of 25%, 30% and 38%, maximal injection time of 140 ms, an AGC set to 800% and a resolution of 60k). The Q-Exactive plus Mass Spectrometer was operated in a data-dependent mode automatically switching between the acquisition of a single Orbitrap MS scan (375–1400 m/z, maximal injection time of 50 ms, an Automated Gain Control (AGC) set to a maximum of $3^*10^6$ ions and a resolution of 70k) and up to 15 Orbitrap MS/MS HCD scans of precursors (Stepped NCE of 28%, 30% and 35%, a maximal injection time of 100 ms, an AGC set to a maximum of $2^*10^5$ ions and a resolution of 17.5k).

## Proteomic analysis

Protein searches were undertaken within MaxQuant (1.5.3.30, v1.6.3.4, or v1.6.17.0.) [55]. Depending on the samples searches undertaken using combination of the human proteome (Uniprot Accession: UP000005640), the Escherichia coli O127:H6 (strain E2348/69) proteome (Uniprot: UP000001521) or a custom database containing the predicted sequence of the RIPK1-Death Domain, 6xHis-tagged FADD, $FAS_{DD}$, $TRAILR2_{DD}$, Flag-$TNFR1_{DD}$, Flag-TRADD or NleB2. Searches were undertaken using "gluC" enzyme specificity for RIPK1-DD samples, A "lys-C" enzyme specificity for NleB2 and "Trypsin" enzyme specificity for all other samples. Carbamidomethylation of cysteine was allowed as a fixed modification as well as the variable modifications of oxidation of methionine, Arg-GlcNAcylation ($H_{13}C_8NO_5$; 203.0793Da to Arginine) and Arg-Glucosylation ($H_{10}O_5C_6$; 162.052 Da to Arginine). To enhance the identification of peptides between samples, the Match between Runs option was enabled with a precursor match window set to 0.75 minutes and an alignment window of 20 minutes with the label free quantitation (LFQ) option enabled [56]. The resulting data was visualized using ggplot2 [57] within R. The resulting MS data and search results have been deposited to the ProteomeXchange Consortium via the PRIDE [58] partner repository and can be accessed with the dataset identifier PXD021796 (In vitro RIPK1DD and NleB2 glycosylation); PXD025057 (RIPK1 glycosylation infection assays) and PXD025531 (Confirmation of Arg-glucosylation of other death domain proteins).

## Supporting information

**S1 Fig. *In vitro* glycosylation assays of NleB2$_{DXD}$ with RIPK1$_{DD}$.** Deconvoluted intact mass spectra of MBP-RIPK1$_{DD}$ incubated with GST-NleB2 either without sugar donors, or in the presence of one of UDP-GlcNAc, UDP-glucose, UDP-GalNAc, UDP-galactose, UDP-glucuronic acid or GDP-mannose at 50 μM.
(TIF)

**S2 Fig. Short *in vitro* incubation with low concentration of sugar donor.** MBP-RIPK1$_{DD}$ was incubated with GST-NleB2 either without sugar donors, or in the presence of one of UDP-GlcNAc, UDP-glucose, or UDP-galactose at 0.5 μM for only 20 minutes.
(TIF)

**S3 Fig. NleB2 modifies arginine 603 within the death domain of RIPK1. (A)** Peptide isolated from MBP-RIPK1$_{DD}$ showing single and double Arg-glucose modifications. MBP-RIPK1$_{DD}$ was incubated with GST-NleB2 in the presence of 10mM UDP-glucose. **(B)** Intact mass spectra of MBP-RIPK1$_{DD}$ incubated with GST-NleB2 either without sugar donors, or in the presence of one of UDP-GlcNAc, UDP-glucose or UDP-galactose at 10 mM.
(TIF)

**S4 Fig. Extracted ion chromatograms of NleB2 *in vitro* glycosylated-RIPK1$_{DD}$.** The glycosylated and non-glycosylated forms of the Arg603-containing Glu-C peptide NLGKHWKNC ARKLGFTQSQIDE from MBP-RIPK1$_{DD}$ observed after incubation of GST-NleB2 without sugars **(A)**, or in the presence of 10mM UDP-GlcNAc **(B)**, UDP-glucose **(C)** or UDP-galactose **(D)** are shown.
(TIF)

**S5 Fig. NleB2 does not modify RIPK1$_{DD(R603A)}$. (A)** Deconvoluted intact mass spectra of MBP-RIPK1$_{DD(R603A)}$ (Full-length expected average mass 52996 Da) incubated with GST-NleB2 either without sugar donors, or in the presence of one of UDP-GlcNAc, UDP-glucose or UDP-galactose at 10 mM. **(B)** Immunoblots of *in vitro* glycosylation assays. MBP-RIPK1$_{DD}$ or MBP-RIPK1$_{DD(R503A)}$ were incubated in the presence of 25 μM UDP-GlcNAc either alone, or with GST-NleB2. Proteins were probed with anti-ArgGlcNAc, or anti-MBP and anti-GST as controls. Representative of at least 3 experiments. **(C-E)** Extracted ion chromatograms of Glu-C digested MBP-RIPK1$_{DD(R603A)}$ after incubation with GST-NleB2 in the presence of 10mM UDP-GlcNAc **(C)**, UDP-glucose **(D)** or UDP-galactose **(E)**.
(TIF)

**S6 Fig. NleB2 glucosylates FADD and TNFR1. (A)** Peptide isolated from His-FADD showing hexose modification of Arg117. His-FADD was incubated with GST-NleB2 in the presence of 10mM UDP-glucose *in vitro*. **(B)** Peptides isolated from Flag-TNFR1$_{DD}$ showing hexose and HexNAc modification of Arg376. Flag-TNFR1$_{DD}$ was immunoprecipitated from HEK293T cells co-expressing GFP-NleB2.
(TIF)

**S7 Fig. *In vitro* glycosylation assays of NleB2$_{S252G}$, NleB1 and NleB1$_{G255S}$ with RIPK1$_{DD}$.** Deconvoluted intact mass spectra of MBP-RIPK1$_{DD}$ incubated with GST-NleB2$_{S252G}$ **(A)**, GST-NleB1 **(B)** or GST-NleB1$_{G255S}$ **(C)** in the presence of either UDP-GlcNAc or UDP-glucose at 50 μM.
(TIF)

**S8 Fig. *In vitro* sugar donor competition assays for NleB2$_{S252G}$, NleB1 and NleB1$_{G255S}$.** Deconvoluted intact mass spectra of *in vitro* sugar donor competition assays. MBP-RIPK1$_{DD}$ was incubated without sugar donors, or in the presence of 25 μM UDP-GlcNAc and UDP-glucose with either GST-NleB$_{2S252G}$, GST-NleB1 or GST-NleB1$_{G255S}$.
(TIF)

**S9 Fig. Kinetic analysis of NleB1 and NleB2 derivatives in the presence of UDP-GlcNAc and UDP-glucose. (A)** Michaelis-Menten kinetics for NleB1 and UDP-GlcNAc as measured using UDP-Glo assay. UDP release was measured after a 30 minute reaction of 150 nM

GST-NleB1 in the presence of titrated concentrations of UDP-GlcNAc alone, or in the presence of UDP-GlcNAc and 1 μM MBP-RIPK1. The mean relative light units (RLU) detected from two replicates is shown with error bars representing standard deviation. **(B)** Michaelis-Menten kinetics for NleB1, NleB2 and derivatives in the presence of UDP-GlcNAc or UDP-glucose as observed using UDP-Glo assays. UDP release was measured after a 30 minute reaction of 150 nM GST-NleB1, GST-NleB2 or derivatives in the presence of 1 μM MBP-RIPK1 and titrated concentrations of either UDP-GlcNAc or UDP-glucose. The mean relative light units (RLU) detected from three replicates is shown with error bars representing standard deviation. **(C)** Vmax and Km values calculated from the data in **(B)** using the non-linear regression fit Michaelis-Menten equation in GraphPad Prism. Values shown are ± standard deviation.

(TIF)

**S10 Fig. Arg-hexose auto-modification of purified NleB2. (A)** Peptide isolated from Lys-C digest of GST-NleB2 showing hexose modification of Arg140 in NleB2. **(B)** Alignment of NleB2 and NleB1 from EPEC O127:H6 strain E2348/69, NleB from *Citrobacter rodentium* strain ICC168 and SseK1, SseK2 and SseK3 from *Salmonella enterica* serovar Typhimurium strain SL1344. Arrow indicates arginine 140 within NleB2. Alignment was performed using ClustalW and visualised using ESPript. **(C)** Extracted ion chromatograms of GST-NleB2 showing the glycosylated and non-glycosylated forms of the Arg140-containing Lys-C peptide LSDIYHDIICEQRLRTEDK.

(TIF)

**S1 Table. Identification of NleB2 mediated modifications in death domain proteins. (A)** Maxquant protein identification information for FADD *in vitro* glycosylation assays. For each sample the summed ion intensity, number of MS/MS events, score, sequence coverage, LFQ values and *in vitro* condition information are provided. **(B)** Maxquant peptide identification information for FADD *in vitro* glycosylation assays. For peptides identified within *in vitro* assays the peptide sequences, modification status, protein name, ion intensity, number of MS/MS events, score, data file of the best identification and *in vitro* condition information are provided. **(C)** Maxquant protein identification information for FasDD and TRAIL2 *in vitro* glycosylation assays. For each sample the summed ion intensity, number of MS/MS events, score, sequence coverage, LFQ values and *in vitro* condition information are provided. **(D)** Maxquant peptide identification information for FasDD and TRAIL2 *in vitro* glycosylation assays. For peptides identified within *in vitro* assays the peptide sequences, modification status, protein name, ion intensity, number of MS/MS events, score, data file of the best identification and *in vitro* condition information are provided. **(E)** Maxquant protein identification information for for pFlag-TNFR1DD / pFlag-TRADD and pGFP-NleB2 co-transfection assays. For each sample the summed ion intensity, number of MS/MS events, score, sequence coverage, LFQ values and co-transfection information are provided. **(F)** Maxquant peptide identification information for pFlag-TNFR1DD / pFlag-TRADD and pGFP-NleB2 co-transfection assays. For peptides identified within co-transfection assays the peptide sequences, modification status, protein name, ion intensity, number of MS/MS events, score, data file of the best identification and *in vitro* condition information are provided.

(XLSX)

**S2 Table. Arginine-glucosylation of Flag-RIPK1$_{DD}$ during EPEC infection of transfected HEK293T cells.** Flag-RIPK1 peptides identified from immunoprecipitations performed on EPEC-infected and transfected HEK293T cells.

(XLSX)

## Acknowledgments

We gratefully acknowledge Lorraine O'Reilly and Andreas Strasser for the gift of FasL. We thank the Melbourne Mass Spectrometry and Proteomics Facility of The Bio21 Molecular Science and Biotechnology Institute at The University of Melbourne for the support of mass spectrometry analysis.

## Author Contributions

**Conceptualization:** Cristina Giogha, Nichollas E. Scott, Jaclyn S. Pearson, Elizabeth L. Hartland.

**Data curation:** Cristina Giogha, Nichollas E. Scott.

**Formal analysis:** Cristina Giogha, Nichollas E. Scott, Elizabeth L. Hartland.

**Funding acquisition:** Cristina Giogha, Nichollas E. Scott, Jaclyn S. Pearson, Elizabeth L. Hartland.

**Investigation:** Cristina Giogha, Nichollas E. Scott, Tania Wong Fok Lung, Georgina L. Pollock, Marina Harper, Ethan D. Goddard-Borger, Jaclyn S. Pearson.

**Methodology:** Cristina Giogha, Nichollas E. Scott, Tania Wong Fok Lung, Georgina L. Pollock, Marina Harper, Ethan D. Goddard-Borger, Jaclyn S. Pearson.

**Project administration:** Cristina Giogha, Nichollas E. Scott, Jaclyn S. Pearson.

**Resources:** Tania Wong Fok Lung, Georgina L. Pollock, Marina Harper, Ethan D. Goddard-Borger.

**Supervision:** Cristina Giogha, Nichollas E. Scott, Jaclyn S. Pearson, Elizabeth L. Hartland.

**Validation:** Cristina Giogha, Nichollas E. Scott.

**Visualization:** Cristina Giogha, Nichollas E. Scott, Ethan D. Goddard-Borger, Elizabeth L. Hartland.

**Writing – original draft:** Cristina Giogha, Elizabeth L. Hartland.

**Writing – review & editing:** Cristina Giogha, Nichollas E. Scott, Elizabeth L. Hartland.

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
