## [Decision Letter · Decision Letter 0]

3 Feb 2021

Dear Prof Hartland,

Thank you very much for submitting your manuscript "NleB2 from enteropathogenic Escherichia coli is a novel arginine-glucose transferase effector" for consideration at PLOS Pathogens. As with all papers reviewed by the journal, your manuscript was reviewed by members of the editorial board and by several independent reviewers. The reviewers appreciated the attention to an important topic. Based on the reviews, we are likely to accept this manuscript for publication, providing that you modify the manuscript according to the review recommendations.

All the reviewers highly evaluated this study. Please consider the additional experiments that Reviewer 3 suggested, as the data would add value to the manuscript.

Sincerely,

Tomoko Kubori, Ph.D.

Associate Editor

PLOS Pathogens

Brian Coombes

Section Editor

PLOS Pathogens

Kasturi Haldar

Editor-in-Chief

PLOS Pathogens

orcid.org/0000-0001-5065-158X

Michael Malim

Editor-in-Chief

PLOS Pathogens

orcid.org/0000-0002-7699-2064

All the reviewers highly evaluated this study. Please consider the additional experiments that Reviewer 3 suggested, as the data would be great input to the story.

Reviewer Comments (if any, and for reference):

Reviewer's Responses to Questions

**Part I - Summary**

Reviewer #1: The manuscript is interesting and reader friendly, as it’s easy to follow. Very good article overall. Note that I am not qualified to evaluate the quality of the MS analysis, but I assume that it is OK. A major question that remain open is related to the function of native NleB2 in the context of infection. What is the phenotype of NleB2 mutant, if any? Is this phenotype, or lack of it, related simply to the native expression levels? I hope that the authors can provide answers and if not they should discuss this issue. Also. it would be nice, although not essential, to have the Km towards the different substrates of NleB1, NleB2, NleB1 G255S and NleB2 S252G. Finally, Figure 7B, the level of NleB1G255S is high comparing to pNleB1. Please comment on this. Is this a stability issue?

Reviewer #2: In this paper, Giogha and colleagues have convincingly characterized a novel bacterial arginine-glucose transferase. Additional biochemical analysis, including kinetic analysis of this enzyme for sugar preferences and usages, is anticipated to provide other critical information. Furthermore, a molecular tool specifically detecting host glucosylated substrates is also expected to help identify this enzyme's host substrates (if there are more) and characterize the roles of this effector in the context of infection. Nonetheless, the data presented in this paper sufficiently and convincingly support the finding of a new arginine-glucose transferase enzyme. My comments and suggestions are as follows.

1. Is the temporal expression regulation of NleB1 and B2 during infection known? Are both effectors expressed and translocated to host cells at a similar level during the course of infection? Or Do those differ in different conditions? Relevant information can be discussed in lines 327-338. If appropriate, this discussion can be extended by comparing and contrasting with SseK1, K2, and K3’s expressions, translocations, and roles.

2. In lines 229-232, the band of approximately 28 kDa is pointed out. What is your interpretation? Is this an auto-glycosylated band? It seems that the authors think this is an important observation, but a relevant description is omitted in the paper.

3. In lines 249-251, this is a confusing sentence. This is associated with many bands in the blot shown in Fig 7B. Please consider rephrasing the sentence and/or improving the figure panel and legend.

4. Figure legends need to be extended to explain figure panels better. For instance, Fig. 1A acronym DDO and QDO are used without full names. In Fig. 1B, TNF-a needs to be used instead of TNF since there are different forms. The acronym SEM also needs to be spelled out in Fig. 1.

5. Fig. 5B legend does not describe color codes of the sugar moiety; for instance, indicating color codes for nitrogen (blue) and oxygen (red) will help orient the N-acetyl group's location.

6. Fig. 6A and 6B descriptions are the same. Exposure time difference should be indicated in Fig. 6B legend and lines 229-232.

7. The authors can consider showing individual data points in bar graphs (e.g., Fig. 1B, 7A).

8. Supplementary figure panels S4A, B, C, & D, S5C, D, & E, and S8C are cropped, and no graphs are shown.

9. At line 183, please confirm whether 10 mM UPD-sugar is correct since, in other experiments, 50 µM or 25 µM sugars are indicated (e.g., Figure legends to Figs. 3-6).

Reviewer #3: In this paper, Giogha et. al. present evidence that the T3SS effector NleB2 from EPEC and EHEC transfers glucose from UDP-glucose to Arg residues on the Death Domain (DD) of RIPK1. Previous studies have shown that the NleB2 paralog, NleB1 transfers GlcNAc from UDP-GlcNAc to Arg residues on DD containing proteins involved in the host immune response. Moreover, NleB2 was tested in these studies and shown to have low levels of Arg-GlcNAcylation of TRADD. Here the authors identified an interaction between NleB2 and the DD domains of RIPK1 and TNFR1 using a yeast 2 hybrid screen and convincingly show that NleB2 (but not a catalytically inactive mutant), like NleB1, inhibits TNFa-induced NFkB activation. Global Arg-GlcNAcylation was observed in lysates from cells overexpressing NleB1 using an anti-Arg GlcNAcylation antibody but not in cells overexpressing NleB2. Infection experiments suggest that NleB2 is not involved in Arg-GlcNAcylation of host proteins during EPEC infection. The authors go on to show that NleB2 utilizes UDP-glucose to Arg-glucosylate RIPK1 on Arg603 within the DD of RIPK1. Structure guided mutagenesis of NleB2 based on the active site architecture of an NleB homolog, revealed that Ser 252 in NleB2 dictates specificity between NleB1 and NleB2 for UDP-GlcNAc and UDP glucose, respectively.

Overall, this is a nice study that adds to our understanding of Arg-glucosylation and EPEC/EHEC pathogenesis. I recommend publication provided the authors can address the following points.

**Part II – Major Issues: Key Experiments Required for Acceptance**

Reviewer #1: none

Reviewer #2: NA

Reviewer #3: 1. Proper kinetic measurements should really be done to compare the Km for UDP-glucose vs UDP-GlcNAc (and the other UDP-sugars used in the paper).

2. It should be shown that NleB2 mediates glucosylation of proteins during infection. If one were to IP overexpressed RIPK1 from infected cells, do you observe Arg-glucosylation by MS? Does this go away in NleB2 KO cells?

3. The authors should test whether other potential DD-containing substrates are glucosylated by NleB2. Eg. TNFR1, TRADD, etc.

**Part III – Minor Issues: Editorial and Data Presentation Modifications**

Reviewer #1: in the summary

Reviewer #2: Minor issues are indicated above.

Reviewer #3: (No Response)

PLOS authors have the option to publish the peer review history of their article (what does this mean?). If published, this will include your full peer review and any attached files.

Reviewer #1: No

Reviewer #2: No

Reviewer #3: No
---

## [Decision Letter · Decision Letter 1]

20 May 2021

Dear Prof Hartland,

We are pleased to inform you that your manuscript 'NleB2 from enteropathogenic Escherichia coli is a novel arginine-glucose transferase effector' has been provisionally accepted for publication in PLOS Pathogens.

Best regards,

Tomoko Kubori, Ph.D.

Associate Editor

PLOS Pathogens

Brian Coombes

Section Editor

PLOS Pathogens

Kasturi Haldar

Editor-in-Chief

PLOS Pathogens

orcid.org/0000-0001-5065-158X

Michael Malim

Editor-in-Chief

PLOS Pathogens

orcid.org/0000-0002-7699-2064

The authors addressed all the concerns raised by the reviewers.

Reviewer Comments (if any, and for reference):

Reviewer's Responses to Questions

**Part I - Summary**

Reviewer #3: The authors have done a good job addressing my concerns, I recommend publication.

**Part II – Major Issues: Key Experiments Required for Acceptance**

Reviewer #3: (No Response)

**Part III – Minor Issues: Editorial and Data Presentation Modifications**

Reviewer #3: (No Response)

PLOS authors have the option to publish the peer review history of their article (what does this mean?). If published, this will include your full peer review and any attached files.

Reviewer #3: No

---

## [Editor Report · Acceptance letter]

11 Jun 2021

Dear Prof Hartland,

We are delighted to inform you that your manuscript, "NleB2 from enteropathogenic *Escherichia coli* is a novel arginine-glucose transferase effector ," has been formally accepted for publication in PLOS Pathogens.

Best regards,

Kasturi Haldar

Editor-in-Chief

PLOS Pathogens

orcid.org/0000-0001-5065-158X

Michael Malim

Editor-in-Chief

PLOS Pathogens

orcid.org/0000-0002-7699-2064